# Terrorism group prediction using feature combination and BiGRU with self-attention mechanism

Mohammed Abdalsalam[1,2,3], Chunlin Li[1], Abdelghani Dahou[4] and Natalia Kryvinska[5]

[1] School of Computer Science and Artificial Intelligence, Wuhan University of Technology, Wuhan, Hubei, China
[2] School of Computer Science and Information Technology, Sudan University of Science and Technology, Khartoum, Khartoum, Sudan
[3] School of Computer Science and Information Technology, University of the Holy Quran and Islamic Sciences, Khartoum, Khartoum, Sudan
[4] LDDI Laboratory, Faculty of Science and Technology, University of Ahmed DRAIA, Adrar, Adrar, Algeria
[5] Information Systems Department, Faculty of Management, Comenius University in Bratislava, Bratislava, Bratislava, Slovakia

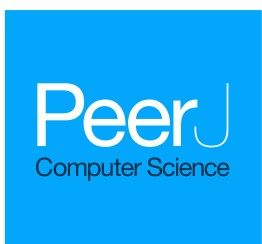

Corresponding author
Mohammed Abdalsalam, 79834@whut.edu.cn

## ABSTRACT

The world faces the ongoing challenge of terrorism and extremism, which threaten the stability of nations, the security of their citizens, and the integrity of political, economic, and social systems. Given the complexity and multifaceted nature of this phenomenon, combating it requires a collective effort, with tailored methods to address its various aspects. Identifying the terrorist organization responsible for an attack is a critical step in combating terrorism. Historical data plays a pivotal role in this process, providing insights that can inform prevention and response strategies. With advancements in technology and artificial intelligence (AI), particularly in military applications, there is growing interest in utilizing these developments to enhance national and regional security against terrorism. Central to this effort are terrorism databases, which serve as rich resources for data on armed organizations, extremist entities, and terrorist incidents. The Global Terrorism Database (GTD) stands out as one of the most widely used and accessible resources for researchers. Recent progress in machine learning (ML), deep learning (DL), and natural language processing (NLP) offers promising avenues for improving the identification and classification of terrorist organizations. This study introduces a framework designed to classify and predict terrorist groups using bidirectional recurrent units and self-attention mechanisms, referred to as BiGRU-SA. This approach utilizes the comprehensive data in the GTD by integrating textual features extracted by DistilBERT with features that show a high correlation with terrorist organizations. Additionally, the Synthetic Minority Over-sampling Technique with Tomek links (SMOTE-T) was employed to address data imbalance and enhance the robustness of our predictions. The BiGRU-SA model captures temporal dependencies and contextual information within the data. By processing data sequences in both forward and reverse directions, BiGRU-SA offers a comprehensive view of the temporal dynamics, significantly enhancing classification accuracy. To evaluate the effectiveness of our framework, we compared ten models, including six traditional ML models and four DL algorithms. The proposed BiGRU-SA framework demonstrated outstanding performance in classifying 36 terrorist organizations

responsible for terrorist attacks, achieving an accuracy of 98.68%, precision of 96.06%, sensitivity of 96.83%, specificity of 99.50%, and a Matthews correlation coefficient of 97.50%. Compared to state-of-the-art methods, the proposed model outperformed others, confirming its effectiveness and accuracy in the classification and prediction of terrorist organizations.

# INTRODUCTION

Terrorism poses a significant threat to international peace and security, manifesting in violent acts committed for political purposes, including hijackings, attacks on civilians, and the potential use of chemical or nuclear weapons (*Quashie, 2023*). Over the past two decades, terrorism has expanded, with fighters, funds, and weapons flowing increasingly between regions and continents. This expansion has facilitated new alliances between terrorist groups and organized crime, including pirates. Terrorism fundamentally denies and undermines human rights, perpetuating a cycle of denial and destruction.

Global terrorist incidents, such as the September 11 attacks in the United States, the Paris attacks, and the London bombings, have had lasting impacts on societies worldwide. These events have reshaped geopolitics and international security dynamics, prompting global reflection on terrorism and security. While these tragedies remain ingrained in collective memory, they have also fostered resilience, unity, and efforts toward peace and security. The aftermath of such incidents has catalyzed international endeavors to combat terrorism, promote intelligence-sharing, and address the root causes of extremism. Acts of terrorism, including hijackings, kidnappings, and assaults on civilians, have garnered significant attention and concern. The international community acknowledges the urgency of confronting this threat and is committed to preventing and combating terrorism.

Mechanisms and methodologies have been developed to address negative phenomena in human behavior, including political extremism, violence, and terrorism (*Abdalsalam et al., 2024*). Traditional analytical methods often struggle with the complexity and volume of terrorism-related data (*Hariri, Fredericks & Bowers, 2019*). Artificial intelligence (AI), particularly machine learning (ML) and deep learning (DL), presents a promising solution to these challenges (*Jeong, 2020*). This study explores the application of ML and DL techniques to predict and identify organizations behind terrorist attacks, aiming to deepen our understanding of terrorist activities and enhance counterterrorism efforts. ML and DL offer the capability to analyze vast amounts of heterogeneous data, including textual, temporal, and spatial information, extracting meaningful patterns and insights. The study investigates various ML and DL approaches, such as bidirectional recurrent neural networks, attention mechanisms, and ensemble learning, to address the multifaceted nature of terrorist data (*Hassani et al., 2021*).

The Global Terrorism Database (GTD) serves as a comprehensive repository of terrorist incidents worldwide, documenting intricate event details, including the responsible group names (Gname) (*Jović et al., 2023*). The frequency of these groups varies widely, contributing to data imbalance and complicating accurate attribution. This imbalance favors the majority class in classifier accuracy, hindering the understanding of minority class characteristics. Moreover, techniques for binary classification may not directly apply to multi-classification scenarios.

The mathematical problem can be formulated as follows: Let $D$ represent the GTD dataset, consisting of records of terrorist incidents worldwide. Each incident is characterized by features, including event details and the responsible groups for acts of terror. The dataset $D$ is highly imbalanced due to many incidents attributed to "unknown" groups. It can be represented as:

$$D = \{(x_i, y_i)\}_{i=1}^{N} \tag{1}$$

where $x_i$ represents the features of the $i$-th incident, and $y_i$ denotes the corresponding responsible group ('Gname'). The set of possible group labels is denoted by $Y$.

The challenge is to develop a predictive model $f : X \rightarrow Y$ accurately identifying the responsible group $y_i$ for a given incident $x_i$. To address the issue of data imbalance in the group names within the GTD, we employed the Synthetic Minority Over-sampling Technique (SMOTE) combined with Tomek links (SMOTE-T) for comprehensive sampling (*Hasan et al., 2024*). This approach combines oversampling and undersampling strategies to boost the efficiency of the classifier model. The SMOTE method generates new samples for the minority class to oversample it, while the Tomek links technique identifies and eliminates samples from the majority class. This process achieves a more equitable distribution of demographic characteristics, thereby optimizing the predictive model $f$ to minimize classification errors.

$$\min_{f} \frac{1}{N} \sum_{i=1}^{N} L(y_i, f(x_i)) \tag{2}$$

where $L$ measures the disparity between predicted and true groups. The model's effectiveness depends on navigating dataset complexities, including sparse entries and evolving terrorist tactics.

The motivation driving this research stems from the imperative need for precise and efficient identification of terrorist groups responsible for attacks. Leveraging ML and DL algorithms on the GTD dataset, we aim to construct a predictive framework capable of accurately classifying and identifying the names of terrorist groups involved in various attacks. Such a framework holds the potential to bolster counter-terrorism endeavors and elevate security measures significantly.

The proposed framework addresses the classification and prediction of terrorist groups responsible for attacks (Gname) by harnessing features from GTD, focusing on textual features (Summary) that encapsulate the essence of each terrorism event. The framework encompasses several pivotal steps to enhance the accuracy of terrorism group detection.

Initially, exploratory data analysis (EDA) is conducted to gain insights into the dataset (*Indrakumari, Poongodi & Jena, 2020*), followed by preprocessing to handle missing values prevalent in the GTD. Based on the Pearson correlation coefficient (PCC) and Normalized Mutual Information (NMI) (*Choi & Kim, 2024*), correlation analysis is applied to categorical and numerical features to identify relationships. The Summary attribute, containing textual information about each terrorism event, undergoes preprocessing to ensure optimal feature extraction. DistilBERT, a state-of-the-art language model, is employed for feature extraction, capturing nuanced information embedded within the summary texts. Given the unbalanced nature of the Gname attribute in the GTD, with over 3,000 unique groups, the framework employs a selection criterion based on attack frequency. Groups with attack frequencies of at least 500 are chosen to ensure a sufficient sample size for analysis, mitigating the imbalanced nature of the target variable. Subsequently, the SMOTE-T are applied to balance the dataset. SMOTE-T effectively combines undersampling and oversampling strategies to address class imbalances, generating synthetic samples for underrepresented groups and removing noisy samples using Tomek links. The framework then leverages the bidirectional gated recurrent unit with self-attention (BiGRU-SA) model for classification and prediction tasks. The BiGRU-SA model capitalizes on the bidirectional architecture of the gated recurrent unit (GRU), capturing temporal dependencies within sequential data. Additionally, self-attention mechanisms enhance the model's ability to focus on relevant information within the input sequence, further improving classification performance. Combining data preprocessing techniques, sampling methods, and advanced neural network models enhances the framework's ability to handle complex and imbalanced datasets effectively, leading to more robust and reliable results in terrorism group classification tasks. The main contributions of this study can be summarized as follows:

- In this study, novel methods for preprocessing the GTD are introduced to address challenges posed by missing values and noisy data. Failure to preprocess the GTD can significantly impact the accuracy and generalization ability of predictive models. To mitigate this issue, we conduct an in-depth analysis of feature distribution and missing data patterns, removing features with substantial gaps. Leveraging interconnections between features, we employ diverse approaches, including web crawling, to impute missing values and generate refined features. Furthermore, we utilize the covariance matrix to identify features exhibiting strong correlations and eliminate those with high information entropy.

- To address the potential challenges posed by the high complexity and dimensionality of data, the framework utilizes DistilBERT. DistilBERT is a lighter version of BERT (Bidirectional Encoder Representations from Transformers), designed to retain much of BERT's performance while being more efficient. This choice reflects a careful balance between maintaining analytical depth and managing computational resources effectively.

- The framework incorporates the BiGRU-SA model. This choice underscores the framework's emphasis on capturing temporal dependencies and the importance of

context within the data. Bidirectional gated recurrent unit (BiGRU), with its bidirectional processing, can understand data sequences in both forward and reverse directions, offering a comprehensive view of the data's temporal dynamics.

The remaining set of this article is organized as follows: "Background and Related Works" explores existing literature and studies relevant to our investigation. "Proposed Framework" elaborates on our innovative methodology and approach to identifying terrorist organizations. "Data Preprocessing" describes the setup and design of the experiments implemented in this research. "Feature Engineering" details the sequence of experiments to evaluate our proposed framework. "Experiments" presents the results of our experiments and delves into a comprehensive discussion of the findings. Finally, "Results, Analysis, and Discussion" summarizes the study, addresses its limitations, and proposes avenues for future research.

## BACKGROUND AND RELATED WORKS

The literature review underscores the growing role of ML and DL in bolstering counter-terrorism endeavours through their capacity to analyze vast datasets effectively. Recent studies have explored the application of ML techniques to forecast and understand terrorist activities. Accordingly, an integrated DL framework was introduced to incorporate the contextual background of previous attack locations, social network dynamics, and historical behaviors of individual terrorist groups (*Jiang et al., 2023*). This study aimed to uncover behavioral patterns among terrorist groups, surpassing conventional base models across various spatiotemporal resolutions. Additionally, the model demonstrated the capability to forecast future targets of active terrorist groups, identifying high-risk areas and providing sequence-based attack-related insights for specific groups. This study highlights the potential of combining DL methodologies with multi-scalar data to offer groundbreaking insights into terrorism and other organized violent crimes.

A Bayesian neural network (BNN) was used to predict characteristics of terrorist attacks in Nigeria (*Ogundunmade & Adepoju, 2024*). The study assessed various activation functions and training datasets, finding that the hyperbolic tangent activation function outperformed others in predicting key variables linked to terrorist attacks in Nigeria.

Another study investigated the spatial prediction of terrorism across Europe by leveraging satellite images and socio-environmental data (*Buffa et al., 2022*). Employing five distinct ML models, they classified the presence or absence of prior attacks within hexagonal-grid cells, highlighting the utility of spatial, ML, and remote sensing methodologies in understanding terrorist behaviors.

AI techniques were employed to visualize and predict potential terrorist attacks *Huamaní, Alicia & Roman-Gonzalez (2020)*. Using decision trees and random forests on GTD data from 1970 to 2018, the study achieved accuracy rates ranging from 75.46% to 90.41% in forecasting terrorism incidents.

A study concentrated on predicting terrorism-prone continents by amalgamating the support vector machine (SVM) and K-nearest neighbor (KNN) into an ensemble ML

model (*Olabanjo et al., 2021*). Utilizing pre-processed GTD data and feature selection techniques, their results underscored the efficacy of hybrid-based feature selection in predicting terrorism locations.

The Tweet-to-Act (T2A) framework was developed for prompt extraction and dissemination of information related to terrorism from Twitter streams (*Iqbal et al., 2021*). T2A used word embedding to translate tweets into a numerically comparable vector space model and employed Word Mover's Distance (WMD) to cluster tweets about the same event based on semantic similarities. The system also incorporated sequence labeling with bidirectional long short-term memory recurrent neural networks (bLSTM-RNN) to efficiently extract pertinent details from tweets. Demonstrating superior performance, T2A outperformed existing methods with an accuracy of 96% and an F1-score of 86.2%, showcasing its effectiveness for real-time monitoring and information extraction in terrorism event detection.

A system for analyzing earthquakes and terrorist acts using Wikipedia and Wikidata was devised (*Zajec & Mladenić, 2022*). Employing an event argument extraction system coupled with semi-supervised learning, they extracted insights from 315 terrorist acts and 913 earthquakes across multiple languages.

A framework for predicting terrorist attacks based on textual features was developed (*Abdalsalam et al., 2021*). Employing three feature extraction techniques and nine ML models, they significantly enhanced prediction accuracy by extracting textual information.

A study focused on terrorism crimes in India, proposing a model to uncover hidden patterns using ML and statistical analysis on GTD data (*Alam et al., 2020*). The model discerned terrorists' preferred targets and weapons, aiding in predicting the success of terrorist attacks.

A framework for classifying and predicting terrorist organizations based on ensemble learning was proposed (*Pan, 2021*). Utilizing GTD data, their quantitative statistical analysis underscored the efficacy of XGBoost and random forest models among the five employed ML models.

The application of social network theory to analyze recurring patterns of terrorist attacks was initiated (*Li et al., 2019*). A terrorist organization network was constructed based on concurrent participation in attacks, facilitating subsequent statistical analyses and community division methods to classify terrorist organizations into 13 categories. The experimental results corroborated the effectiveness and efficiency of the proposed analytical approach.

The synthesis of recent literature highlights a significant body of research focused on using ML and DL techniques in counterterrorism, such as classifying and predicting terrorist attacks. These studies validate the effectiveness of ML in forecasting and understanding terrorist activities, providing invaluable insights for counterterrorism strategies. Despite significant progress through the application of ML and DL models across various aspects, such as attack prediction, radicalization identification, and pattern analysis, the field continues to encounter challenges, especially in classifying and predicting terrorist organizations.

**Table 1 Summary of related works.**

| Reference | Work goal | Dataset used | Method | Limitation |
|---|---|---|---|---|
| Jiang et al. (2023) | Discover behavior patterns of terrorist groups | GTD, BAAD, EDTG, Pattern-Net | Utilizes DL to analyze terrorist group behaviors by considering past attacked locations, social networks, and past actions. | Relies on incomplete data on unidentified terrorist organizations and potential biases in media-sourced attribution. Model effectiveness may be impacted by data quality and completeness. |
| Ogundunmade & Adepoju (2024) | Develop a model for predicting the nature of terrorist attacks in Nigeria | Nigerian Terrorism Incident Dataset (Nigeria Terror Attack Dataset) | Utilizes a Bayesian neural network to predict terrorist attack types. | Complex methodology and reliance on Bayesian inference may present challenges in model interpretation and implementation. Model effectiveness influenced by data quality and representativeness. |
| Buffa et al. (2022) | Predict terrorism in Europe at the sub-national level | GTD combined with remotely sensed data, geospatial features, and population data | Employs various ML models to predict terrorism presence in hexagonal-grid cells. | Grid-cell approach oversimplifies intricate spatial patterns and may fail to account for neighboring pixel values. Dataset reliance on a static temporal snapshot may not capture changes in land use and socio-environmental factors over time. |
| Huamaní, Alicia & Roman-Gonzalez (2020) | Visualize and predict terrorist attacks globally | GTD | Uses classification models like decision trees and random forest to visualize and predict terrorist attacks globally. | Method overlooks pertinent features for terrorism attack types and inadequately addresses missing values and repeated information in the dataset. |
| Olabanjo et al. (2021) | Predict continents susceptible to terrorism | GTD | Proposes an ensemble ML model to predict continents susceptible to terrorism. | Predicts continents rather than individual countries, potentially limiting insights. |
| Iqbal et al. (2021) | Extract terrorist attack-related information using Twitter | Twitter event dataset | Utilizes word embedding and WMD for clustering tweets, and bLSTM-RNN for sequence labeling. | Relies on Twitter as a primary data source, introducing biases and limitations. Effectiveness may vary depending on the volume and nature of Twitter activity. |
| Zajec & Mladenić (2022) | Develop robust event argument extraction systems | Wikipedia articles and Wikidata information | Introduces a semi-supervised methodology for training an event argument extraction system. | Relies on pseudo-labeling and semi-supervised learning, potentially resulting in noisy labels. Overlooks valuable information present in articles written in languages with challenging matching. |
| Abdalsalam et al. (2021) | Classify terrorism attack types | GTD | Applies ML approach with textual features extraction for terrorism attack classification. | Method primarily focuses on classical feature extraction techniques, potentially limiting the model's ability to capture nuanced patterns. Does not address imbalance in terrorism attack type labels. |
| Alam et al. (2020) | Predict terrorist groups most likely to target a nation | GTD | Utilizes data mining and ML techniques to analyze patterns within the GTD. | Focused on classical ML approaches, may restrict model's adaptability to evolving trends in terrorism. Does not explicitly address imbalanced data in the GTD. Relies on historical data, limiting predictive capabilities for future activities. |
| Pan (2021) | Develop a framework for classifying and predicting terrorist organizations | GTD | Utilizes ensemble learning with five different ML models. | Effectiveness of the framework may be specific to chosen ML models. |
| Li et al. (2019) | Analyze the alliance network of terrorist organizations | GTD | Constructs terrorist network graphs and utilizes ML algorithms for network analysis. | Relies on publicly available data and may face challenges in accurately predicting terrorist activities based solely on historical data. |

Table 1 presents the comprehensive advancements and persistent challenges in this domain. Accordingly, the classification and prediction of terrorist organizations remain relatively unexplored. This gap in research may stem from the complex and dynamic nature of terrorist groups, which frequently evolve, change names, merge, or disband, thereby complicating the creation of accurate and current classification and prediction models. These studies utilize diverse algorithms and methodologies, including support vector machines, decision trees, deep neural networks, and statistical approaches. The GTD often serves as the primary data source for these studies, aiming to improve the understanding of terrorism patterns and provide insights for counterterrorism efforts (*Feyyaz, 2020*).

One of the critical challenges researchers encounter is the issue of missing values within the GTD dataset (*Grossman & Pedahzur, 2020*). This issue can significantly compromise the integrity of the data and the accuracy of analytical results. Traditional methods for handling missing data, such as deletion or imputation, might inadvertently alter the dataset, impacting subsequent analyses and model performance. This study emphasizes the importance of advanced data pre-processing methods, especially for managing missing data, to maintain data integrity and ensure a robust foundation for model development. Furthermore, the choice of feature extraction methods critically affects model predictions. Sole reliance on a single feature type only partially captures semantically relevant information, necessitating an integrated approach combining multiple types of information. Imbalances in data categories further hinder model generalization, often biasing predictions towards the majority class. Traditional classifiers prove inadequate for accurately identifying terrorism groups, underlining the necessity for novel methods that can more effectively extract semantic and contextual information.

To address these challenges, we propose a new framework for identifying the Gname. This framework strives to achieve a more balanced and comprehensive understanding of the factors defining terrorist groups, thereby overcoming the mentioned challenges and propelling counterterrorism research forward. The proposed framework represents a sophisticated and thoughtful approach to analyzing text data related to terrorist organizations. Combining advanced natural language processing (NLP) techniques with strategies specifically tailored to ensure data integrity and reduce computational overhead, the framework is well-positioned to extract meaningful insights from complex datasets. This approach utilizes a bidirectional gated recurrent unit (BiGRU) combined with a self-attention mechanism (SA) to improve the model's ability to detect subtle nuances within the data. This approach enhances the accuracy and depth of the analysis and demonstrates a keen understanding of the challenges inherent in processing and interpreting sensitive and nuanced information.

## PROPOSED FRAMEWORK

This work focuses on designing and implementing a comprehensive framework for classifying and predicting the names of terrorist groups responsible for attacks. The goal is to establish a robust system capable of accurately identifying and predicting these groups with high efficacy. Utilizing the GTD as both the training and testing ground, the

framework applies supervised classification techniques to predict the criminal suspects involved in terrorist activities. An initial phase of EDA provides essential insights into the dataset, setting the stage for subsequent preprocessing tasks. These tasks address missing values and implement feature engineering to refine the dataset for analysis (*Zhang, Cao & Romagnoli, 2018*). We then employ feature selection techniques, including the PCC and NMI, to pinpoint features directly linked to Gname, the nomenclature responsible for attacks (*Maria, Akhand & Shimamura, 2022*; *Karell-Albo et al., 2020*; *Wang et al., 2021*). The proposed approach uses three key strategies to enhance feature assessment and combination for predicting Gname. The first strategy leverages textual features extracted from DistilBERT to ascertain their utility in learning Gname. The subsequent strategies focus on feature combination: first, converting all features to text for DistilBERT-based feature extraction, and then using these extracted features as input to the model. The second strategy involves processing categorical feature sets with label encoding, transforming categorical variables into numerical equivalents. To maintain data integrity and ensure analytical accuracy, numerical features are normalized to prevent disproportionately large values from skewing the results. DistilBERT prowess in extracting textual features from summary attributes is harnessed, merging them with related features to create a cohesive feature set. A sophisticated sampling technique, combining the Synthetic Minority Oversampling Technique with Tomek links (SMOTE-T), is deployed to achieve a balanced training set. We design a model using the advantages of the BiGRU. Transformer-based models and their subsequent generation currently lead in many NLP tasks as the state-of-the-art. However, we selected GRUs over multi-head attention mechanisms due to their strengths in tasks requiring sequential data handling, efficient parameter usage, and robustness to noise (*Wen, Zhou & Su, 2022*).

GRUs are simpler, more stable, and less prone to overfitting. BiGRU, specifically, excel at preserving the order of data and capturing dependencies from both past and future contexts, which is crucial for tasks like time series analysis and NLP, where the sequential nature of data is integral to understanding context and meaning (*Li et al., 2022*).

Furthermore, GRUs typically have fewer parameters compared to multi-head attention mechanisms, which results in lower memory usage and reduced demand on computational resources. This efficiency proves especially beneficial in environments with limited computational capacity, such as mobile devices or embedded systems, and is ideal for processing smaller datasets where extensive resource allocation is not feasible (*El Koshiry et al., 2024*). The overarching research framework and its components are illustrated in Fig. 1.

## Dataset

Terrorism databases play a crucial role in understanding and countering terrorist activities, essential for developing effective security measures. Currently, the research community relies on several internationally recognized terrorism databases, listed in Table 2. These include the Global Terrorism Database (GTD), International Terrorism Attributes of Terrorist Events (ITERATE), the RAND Database of Worldwide Terrorism Incidents (RDWTI), and the Database of Terrorism in Western Europe Events Data (TWEED)

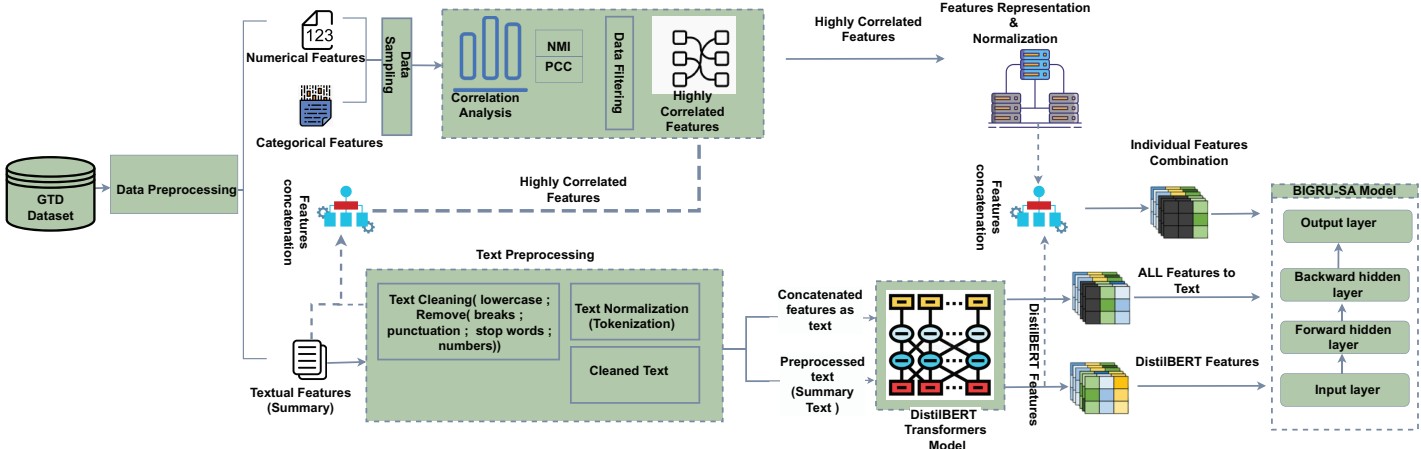

**Figure 1 Proposed framework for classifying and predicting terrorist groups responsible for attacks.** Figure source:© draw.io.

**Table 2 Summary of prominent terrorism databases.**

|  | GTD | ITERATE | RDWTI | TWEED |
|---|---|---|---|---|
| Scope | International + domestic (U.S.A) | International | International + domestic (U.S.A) | Western Europe |
| Time span | 1970–2018 | 1968–2014 | 1968–2009 | 1950–2004 |
| Events | 190,000 approx. | 13,000+ | 40,000+ | 11,245 approx. |
| Variables | 135 variables | 42 | 07 + narrative description | 52 |

(*LaFree, 2019*). These repositories offer invaluable insights into various terrorist incidents, encompassing locations, dates, targets, and specific characteristics. These datasets provide accurate and up-to-date information for formulating effective security plans and strategies. The absence of comprehensive and reliable datasets poses a formidable obstacle to conducting meaningful analyses of global terrorist attacks.

The GTD stands out as a premier database, featuring comprehensive data on international terrorism from the 1970s to 2020. ITERATE, curated by Edward Mickolus, covers the period from 1968 to 1977 and provides detailed insights into terrorist activities and related facets. RDWTI, maintained by the RAND Corporation, records terrorism incidents from 1968 to 2009, offering a comprehensive yet limited perspective due to its lack of updated data. TWEED focuses specifically on terrorism in Western Europe, providing detailed insights into terrorist incidents and unique variables such as state-sponsored activities, enhancing our understanding of terrorism trends in the region. As terrorism research progresses, these databases remain invaluable resources for policymakers, analysts, and researchers seeking deeper insights into global terrorism dynamics.

According to Table 1 and based on the datasets used in state-of-the-art research, the GTD stands out as the optimal choice for ML and DL research. This is due to its comprehensive coverage, regular updates, and recognition as a premier database in

**Table 3 The GTD dataset raw sample.**

| eventid | iyear | imonth | iday | extended | country_txt | resolution | region |
|---|---|---|---|---|---|---|---|
| 198807220004 | 1988 | 7 | 22 | 0 | Pakistan | nan | 6 |
| 201602050022 | 2016 | 2 | 5 | 0 | Nigeria | nan | 11 |
| 200201080001 | 2002 | 1 | 8 | 0 | Indonesia | nan | 5 |
| 201510290019 | 2015 | 10 | 29 | 0 | Thailand | nan | 5 |
| 201501070024 | 2015 | 1 | 7 | 0 | Somalia | nan | 11 |
| 201705260029 | 2017 | 5 | 26 | 0 | Philippines | nan | 5 |
| 200803230007 | 2008 | 3 | 23 | 0 | Iraq | nan | 10 |
| 199702250006 | 1997 | 2 | 25 | 0 | China | nan | 4 |
| 200905240016 | 2009 | 5 | 24 | 0 | India | nan | 6 |
| 200902180018 | 2009 | 2 | 18 | 1 | Afghanistan | 2/19/2009 | 6 |

terrorism research (*Kejriwal, 2021*). Table 3 provides a sample of the dataset, illustrating these attributes for a selection of events. Key features of the GTD dataset include:

1) Detailed records of over 190,000 terrorist attacks, including more than 91,000 bombings, 20,000 assassinations, and 13,000 hostage-taking incidents since 1970 (*Saidi & Trabelsi, 2022*).

2) The most comprehensive unclassified database of terrorist attacks worldwide, with each case containing information on at least 45 attributes, and recent events having more than 130 attributes (*Homolar & Rodríguez-Merino, 2019*).

3) Data accuracy ensured by START through the review of over 4 million news articles and more than 25,000 news sources (*Barnett et al., 2013*).

The GTD is the premier resource for terrorism research, renowned for its comprehensive and authoritative compilation of global terrorism-related information. Developed as an open-source project by the University of Maryland, the GTD covers international terrorism from the 1970s through 2020. Its extensive data includes detailed information such as the date and location of each incident, the weapons used, the targets, the number of victims, and the responsible groups. The GTD currently contains more than 190,000 cases, making it one of the most significant databases in the field. The GTD is distinguished by its commitment to regular updates, ensuring that researchers have access to the most current information on both domestic and international terrorist acts. This dedication to data currency enhances its utility for predictive analysis and strategic planning in counter-terrorism efforts. Unlike many other event databases, the GTD offers in-depth, trustworthy, and open-source data, providing researchers with valuable insights to identify and predict terrorist acts. Each entry in the dataset includes comprehensive details about the incident, the nature of the target, the number of casualties, and specific information about the group or individual responsible for the attack.

The National Consortium for the Study of Terrorism and Response to Terrorism (START) has played a crucial role in disseminating GTD datasets to the public, thereby

fostering a greater understanding of terrorist violence and encouraging participation in terrorism research and resistance efforts. This open-access approach not only facilitates academic research but also empowers policymakers, law enforcement agencies, and other stakeholders in their efforts to combat terrorism effectively.

However, when using the GTD dataset for ML and DL purposes, there are some limitations to consider. These limitations include potential biases or inconsistencies in the data due to variations in reporting and data collection methodologies. Additionally, the dataset has imbalanced categories, where certain types of attacks or regions are overrepresented or underrepresented.

## Data sampling and exploratory data analysis

Data sampling is a common statistical method involving the selection, processing, and analysis of representative data subsets from larger datasets. Exploratory data analysis (EDA) is fundamental for understanding and visualizing datasets, aiming to unveil patterns, trends, and anomalies (*Tariq & Aithal, 2023*; *Mukhiya & Ahmed, 2020*). Integrating data sampling with EDA provides analysts with a powerful method for navigating and extracting insights from complex datasets. This approach allows for efficient selection of representative subsets, which facilitates focused exploration to uncover patterns and trends. In the context of the GTD, characterized by extensive records of terrorist incidents worldwide, this approach becomes indispensable.

In this study, we applied data sampling within the proposed framework to the GTD dataset to overcome challenges posed by its vast size and complexity. Our EDA covers three primary aspects: feature types, spatiotemporal distribution of terrorist attacks, and missing data rates. Firstly, we categorize features into categorical and numerical types, crucial for determining preprocessing and modelling strategies. Secondly, visualization techniques explore the spatiotemporal distribution of terrorist attacks, revealing patterns such as geographical concentration during specific time periods. Lastly, we assess missing data rates to evaluate dataset quality and completeness. Our analysis highlights attributes, particularly detailed perpetrator information, with higher rates of missing data, requiring careful handling during preprocessing to maintain integrity.

### *Feature types and statistical analysis*

Understanding the diverse feature types present in the GTD is paramount for effective preprocessing and subsequent analysis. The GTD dataset encompasses numerical, categorical, and textual features, each requiring tailored preprocessing methods as shown in Table 4.

### *Numerical and categorical features*

The GTD dataset comprises both numerical and categorical features that provide comprehensive information about various aspects of terrorist incidents. Numerical features include quantitative attributes like the number of fatalities, indicating the count of individuals who lost their lives, and the number of injuries, representing individuals who

**Table 4 GTD features description.**

| Attribute | Description |
| --- | --- |
| gname | Name of the perpetrator group |
| iyear | Year when the event occurred |
| imonth/iday | Month and day of the event (1–12)/(1–31) |
| Region/country/state/city | Location (region, country, state, city) where the event occurred |
| Latitude/Longitude | Exact latitude and longitude coordinates of the event location |
| attacktype | Methodology of the attack |
| weapontype | Weaponry utilized in the attack |
| target | Object or person targeted in the attack |
| nkill/nwound | Number of fatalities/injuries resulting from the attack |
| natlty | Nationality of the target |
| suicide | Indicates if the attack was a suicide mission |
| multiple | Indicates if the attack was part of a series of multiple attacks |
| INT_IDEO | Ideological basis of the attack as international or domestic |
| Summary | Attribute provides a brief description of the incident |

sustained injuries during an attack. Additionally, numerical features encompass property damage, quantifying the extent of damage caused, and the year in which the incident occurred. On the other hand, categorical features classify incidents based on specific attributes or characteristics. Examples include country, denoting the location of the incident; region, classifying incidents by geographical regions; attack type, categorizing incidents into different types such as bombing/explosion, armed assault, or assassination; and target type, specifying the type of target attacked, such as private citizens, military, police, or government entities.

### Textual features

Textual features capture qualitative information about terrorist incidents, providing detailed descriptions and contextual information. The most significant textual feature in the GTD dataset is the Summary attribute. The Summary feature encapsulates a brief narrative of each terrorist event, highlighting key details such as:

- **Nature of the incident**: Describes the type of attack and the methods used by perpetrators.
- **Location**: Provides information about where the incident occurred, including specific cities or regions.
- **Targets**: Identifies the targets of the attack, whether they are individuals, government entities, religious institutions, *etc.*
- **Perpetrators**: Describes the group or individual responsible for carrying out the attack.
- **Outcome**: Discusses the consequences of the incident in terms of casualties, injuries, and property damage.

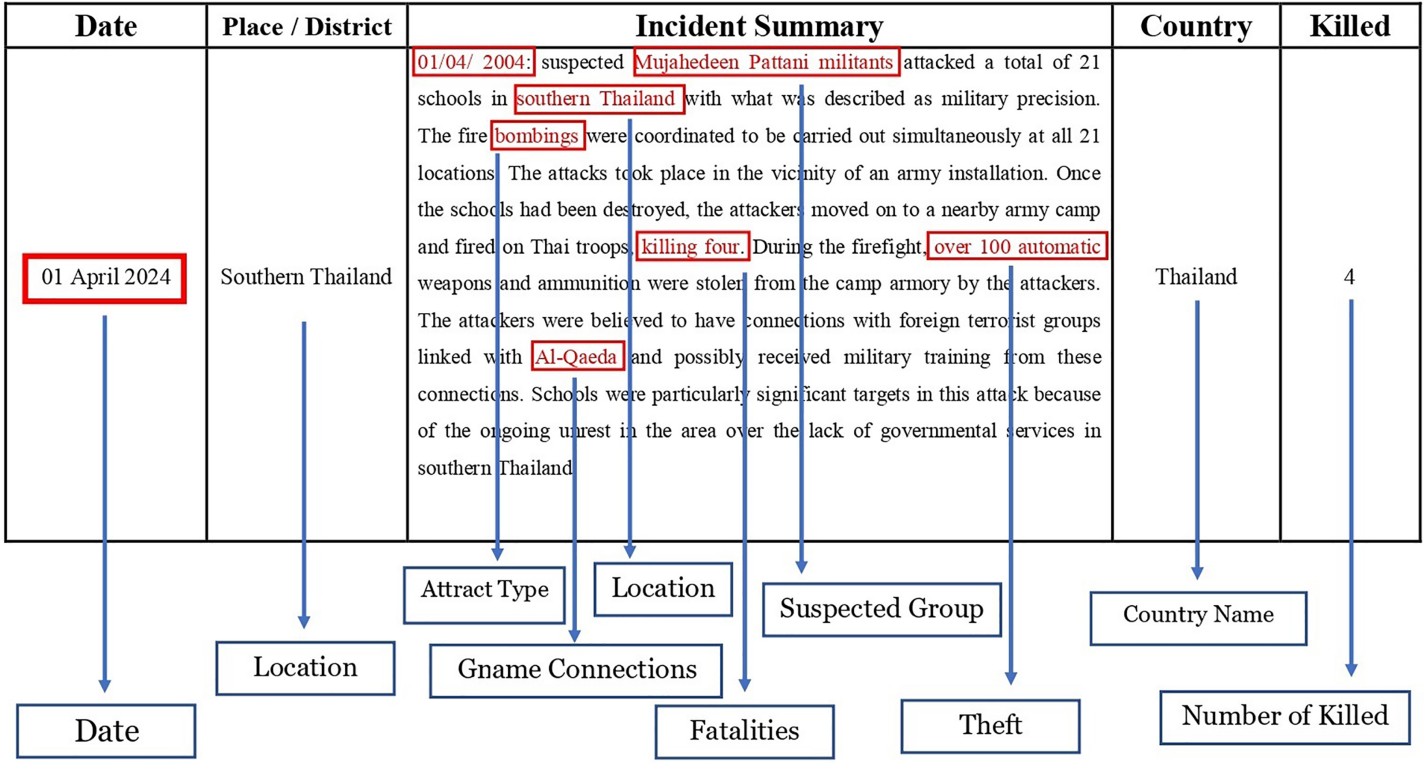

**Figure 2 Summary attributes discovered in collected data.**

### Exploratory data analysis for summary

The summary attribute in GTD is vital for understanding and extracting insights from terrorism incidents. It encompasses essential details such as the time, location, individuals involved, attack process, cause, and other relevant information related to each attack incident.

Figure 2 illustrates event summary information for specific attacks, demonstrating the richness and diversity of the dataset.

The incident summary comprises two main components: the timing of the attack and a detailed description of the attack process. The timing is recorded in the "month/day/year" format, while the attack description includes information such as the attack location, target type, casualty count, and additional details. Figure 2 illustrates the significance of the summary attribute within GTD, highlighting its provision of rich information. The summary encompasses details such as the location, timing, casualties, and the weaponry employed in the attacks. Figure 3 quantitatively presents the frequency of each word, illustrating its prominence in the dataset. Higher frequencies indicate common terms in terrorism incident summaries.

Certain frequently occurring terms in the summary attribute provide valuable insights into the nature of terrorism incidents, such as common attack types, targets. This attribute is crucial for understanding terrorism trends through text mining and NLP techniques. In this study, we applied text preprocessing methods, including tokenization and stop-word

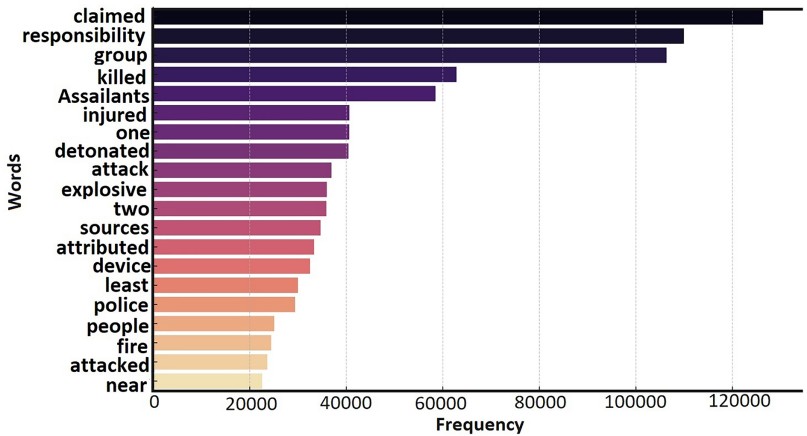

**Figure 3 Most frequently occurring words in summary.**

removal, to extract significant features and semantic understanding—steps that are essential for building effective predictive models and classification algorithms.

Upon analysis, the GTD dataset comprises 135 features per terrorist attack instance, encompassing 54 numerical, 23 categorical, and 58 textual features. These features provide comprehensive insights into various aspects of terrorist incidents, enhancing the dataset's depth and analytical potential. To assess each feature's missing data extent quantitatively, we utilize the missing data rate (*Fiero et al., 2016*), denoted as $\eta$. We calculate this rate using the formula:

$$\eta = \frac{m}{n} \times 100 \tag{3}$$

In Eq. (3), m represents the number of missing values within the feature, while n represents the total number of samples in the dataset. The resulting missing data rate indicates the proportion of missing values for a given feature. This rate reflects the proportion of missing values for each feature, aiding in the analysis of data completeness. Understanding feature types and analyzing missing data distribution are critical steps in ensuring the GTD dataset's readiness for subsequent mathematical modelling tasks. These insights guide effective preprocessing strategies and enhance the dataset's analytical robustness. Visual analysis techniques, including pie charts, further facilitate the examination of missing data distribution. Figure 4 depicts the missing data rate distribution across the GTD dataset, revealing that approximately 56% of features contain missing values.

### Temporal distribution of attacks

Analyzing the temporal distribution of terrorism attacks in the GTD offers profound insights into the evolving patterns and trends of these incidents over time. Examining the dataset allows us to discern significant temporal patterns and better understand the trajectory of terrorism. One effective method to analyze temporal distribution is through visualization, which enables observation of fluctuations in attack frequency over different

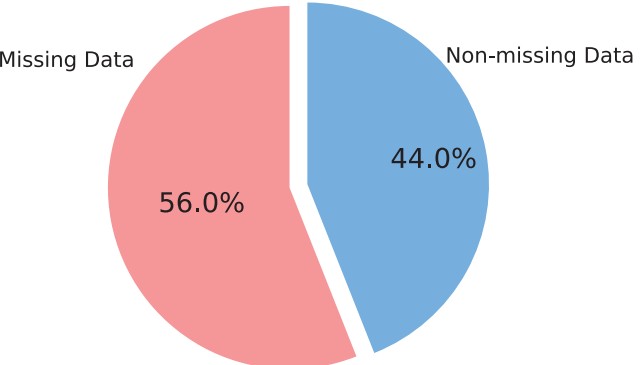

**Figure 4** **An overview of missing values in GTD.**

time periods. Figure 5 illustrates the statistics of the number of terrorist attacks per year, offering a comprehensive overview of attack trends over time.

This visualization enables us to identify spikes or dips in attack frequency, indicating potential patterns or shifts in terrorist activities. For instance, the surge in attacks during the 1980s and 1990s, followed by a decline in the 2000s, suggests dynamic changes in terrorist strategies or counter-terrorism efforts. Furthermore, exploring the temporal distribution of attacks unveils long-term trends or shifts in terrorism dynamics. Notably, the exponential increase in global terrorist attacks from 2011 to 2014, as depicted in Fig. 5, underscores the evolving nature of terrorism and the challenges it poses to global security. The data from GTD, as illustrated in Fig. 5, reveals a staggering 200,000 recorded instances of terrorist attacks between 1970 and 2020, emphasizing the magnitude of the issue. Moreover, the rate of success in recent years, depicted in Fig. 6, underscores the expanding operations of terrorist organizations and their ability to achieve their objectives. Additionally, the rising rates of casualties and fatalities, as evidenced by Fig. 6, highlight the profound impact of terrorist crimes. This visual analysis underscores the urgency for policymakers and security agencies to adapt strategies and allocate resources effectively. By identifying trends and patterns in terrorist activities, stakeholders can enhance their understanding of evolving threats and formulate more effective counter-terrorism measures. Visual representations serve as powerful tools for policymakers, researchers, and security agencies to navigate the complexities of global terrorism.

### Spatial distribution of attacks

Understanding the spatial distribution of terrorist attacks is paramount for devising effective counter-terrorism strategies. By scrutinizing the proximity of attacks to national borders and discerning patterns across neighbouring countries, we can glean insights into the international dimensions of terrorism and potential interconnections between various terrorist groups. It is noteworthy that the spatial distribution of attacks may fluctuate over time due to shifting geopolitical dynamics, conflicts, or counter-terrorism initiatives. The GTD categorizes the world into 12 regions, facilitating an analysis of attack frequencies across different areas. This segmentation enables the identification of hotspots and aids in comprehending the evolving patterns of terrorist threats. Heat maps serve as a suitable

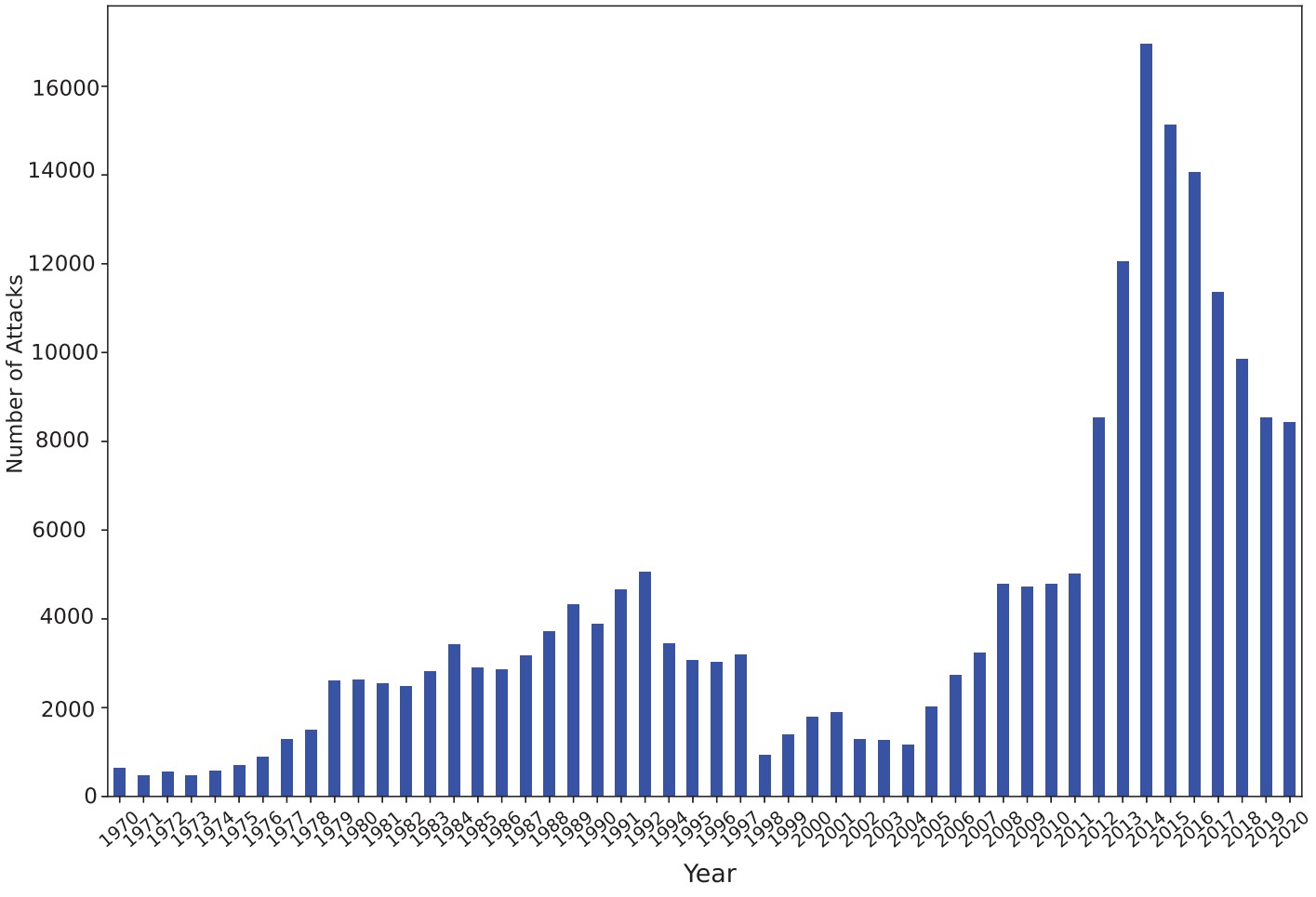

**Figure 5 Statistics of the number of terrorist attacks per year.**

method for visualizing this analysis, offering a graphical representation of attack density or intensity across geographic regions.

Figure 7 portrays the results of the spatiotemporal distribution analysis based on the GTD dataset.

As depicted in Fig. 7, terrorist attacks are predominantly concentrated in four regions: the Middle East and North Africa, South Asia, Sub-Saharan Africa, and Southeast Asia. These regions collectively witness approximately 80% of the total global attacks. Notably, post-2014, there has been a significant decline in the number of attacks across all regions, with the Middle East and North Africa experiencing the most substantial decrease. However, Sub-Saharan Africa, a recognized hub of international terrorist activities, has not witnessed a significant decrease in attack frequency in recent years. Analyzing the spatial distribution of attacks alongside the temporal distribution provides a holistic understanding of terrorism dynamics. It enables the identification of regions with high attack concentrations, exploration of potential cross-border linkages, and detection of changes in attack patterns over time.

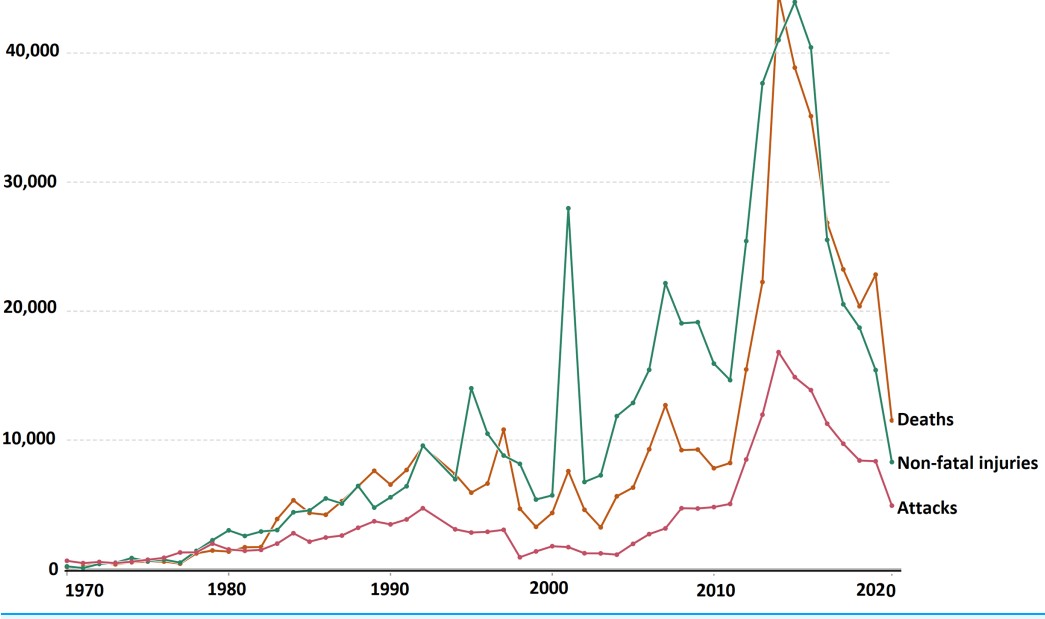

**Figure 6  Frequency yearly killed/casualty and attacks from GTD.**

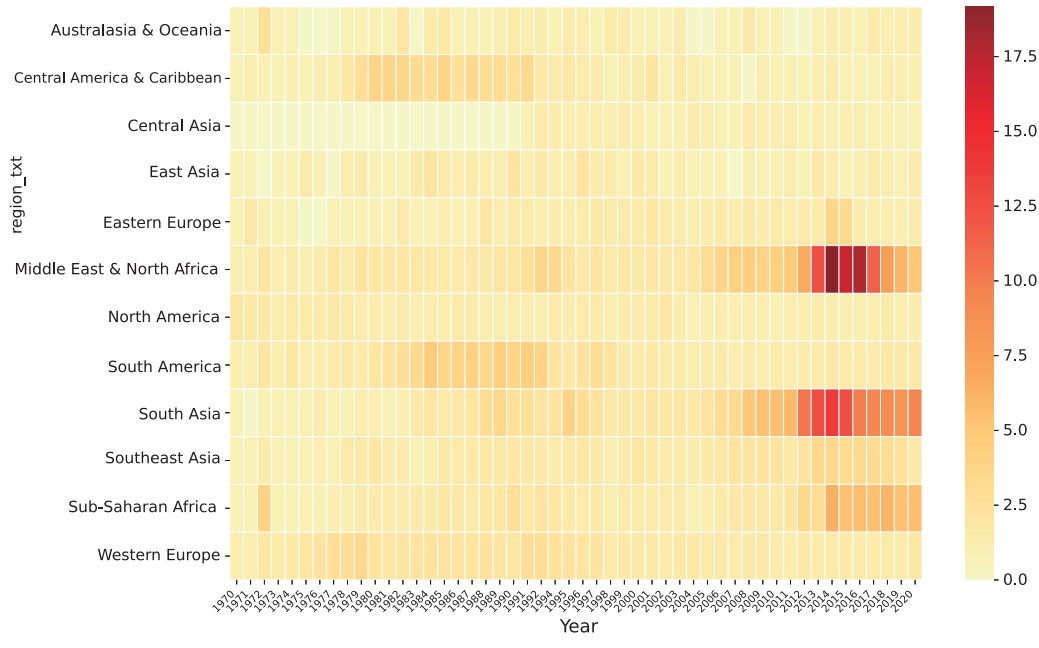

**Figure 7  Time and space distribution map of attacks.**

# DATA PREPROCESSING

Data preprocessing is a crucial phase in ML and DL models where raw data undergoes cleaning, transformation, and formatting to optimize its usability for model training and analysis. Given that raw data often contains inconsistencies, missing values, or irrelevant information, preprocessing ensures that the data is appropriately structured and free from

**Table 5 Fill missing values with fixed values.**

| Feature name | Missing rate | Data type/Filling content |
|---|---|---|
| Nationality of victim | 3.04% | Categorical-String "unknown" |
| Event classification | 83.43% | Integer value 0 |
| Suspected criminal group | 0.31% | Integer value 0 |
| Number of murderers | 10.62% | Integer value-1 |
| Number of murderers caught | 2.72% | Integer value-1 |
| Claimed pattern | 82.90% | Integer value 10 |
| Kidnapped victim | 0.002% | Integer value-9 |
| Is kidnapped victim | 8.35% | String "Unknown" |
| Target victim name | 0.14% | String "Unknown" |
| Target type | 2.50% | Categorical-String "unknown" |
| Weapon type | 1.80% | Categorical-String "unknown" |

anomalies that could adversely affect model performance. Given the prevalence of missing values and noise in the GTD, preprocessing steps are necessary to ensure data reliability and modelling effectiveness. The study initially employs pivot tables and descriptive statistics to evaluate missing values, eliminating features with a missing rate exceeding 90%. To address missing values, the study adopts imputation strategies such as mean, median, or mode imputation for numerical features and categorical imputation for categorical features. These strategies aim to preserve the integrity of the dataset while minimizing the introduction of bias.

- **Fill missing values with fixed values**

The fill missing values is a technique used to replace missing values (nulls) in a dataset with estimated values, minimizing the impact of missing data on subsequent analysis or modelling tasks. Table 5 showcases examples of missing values and their corresponding fixed values for different features in the GTD dataset.

- **Filling missing values based on web crawler**

In our study, to enhance missing values in the "City" and "Country" fields based on latitude and longitude, we used web crawling technology. Web scraping, also known as web crawling, is the process of automatically extracting data from websites using software. It is particularly useful when data is not provided in machine-readable formats like JSON or XML. For this study, a web crawler was employed to address the 4.19% of missing values in the "City Name" column. The crawler leveraged forward and backward geocoding processes to retrieve geolocation information from Google Maps. By simulating interactions with the service, the crawler obtained city and country names from latitude and longitude coordinates, achieving an 84.84% completion rate. If both the city name and coordinates were missing, the crawler attempted to determine the names using available information, with empty returns designated as "unknown."

# FEATURE ENGINEERING

Feature engineering is crucial in developing ML models, involving techniques to enhance model performance and predictive capability (*Nweke et al., 2018*). In the context of the GTD dataset, feature generation and correlation analysis are pivotal for improving model accuracy and eliminating redundant features. Techniques used in this study include selecting pertinent features, transforming variables, extracting new features, amalgamating existing ones, and manipulating data to create variables conducive to analysis and predictive modeling.

## Correlation analysis between features

Correlation quantifies the strength and direction of a linear relationship between features (*Al-Nafjan, 2022*). We employed the PCC for continuous variables and NMI for categorical variables (*Chen et al., 2023*; *Tao et al., 2023*). PCC measures linear relationships among continuous variables, defined as:

$$r = \frac{\mathrm{cov}(X, Y)}{\sigma_X \sigma_Y} = \frac{E(XY) - E(X)E(Y)}{\sqrt{E(X^2) - E^2(X)}\sqrt{E(Y^2) - E^2(Y)}} \tag{4}$$

where $\mathrm{cov}(X, Y)$ denotes the covariance between $X$ and $Y$, and $\sigma_X$ signifies the standard deviation of $X$. Correlation strength ranges: 0–0.3 (low), 0.3–0.8 (moderate), and 0.8–1 (high).

NMI measures similarity between categorical variables, normalized between 0 and 1, calculated as:

$$NMI(X, Y) = \frac{2I(X; Y)}{H(X) + H(Y)} \tag{5}$$

For categorical features, NMI values build a correlation matrix. Steps include:

- **Compute NMI for each pair:** Calculate NMI for every pair of categorical features.
- **Construct a correlation matrix:** Use NMI values to build a matrix showing associations between categorical features.

Table 6 presents the NMI matrix, illustrating relationships among categorical features. Each cell shows NMI values, with higher values indicating stronger associations. Diagonal entries are NaN, as NMI for a feature with itself holds no meaning. The heatmap in Fig. 8 illustrates PCC between numerical features. Darker colors denote stronger correlations, while lighter shades indicate weaker ones. Notable patterns include:

- **Correlations between Attack Statistics and Attack Type:** Higher casualties are associated with attacks targeting civilians.
- **Correlations between Attack Type and Target Nationality:** Specific attack types align with certain target nationalities, indicating targeted patterns.
- **Correlations between Target Characteristics and Attack Type:** The target type influences the choice of attack method, revealing strategic aspects.

**Table 6 NMI matrix for a subset of categorical features.**

|  | country_txt | region_txt | prov-state | city | location |
|---|---|---|---|---|---|
| country_txt | NaN | 0.6907 | 0.7453 | 0.5710 | 0.2748 |
| region_txt | 0.6907 | NaN | 0.4814 | 0.3520 | 0.1895 |
| prov-state | 0.7453 | 0.4814 | NaN | 0.7715 | 0.3613 |
| city | 0.5710 | 0.3520 | 0.7715 | NaN | 0.4277 |
| location | 0.2748 | 0.1895 | 0.3613 | 0.4277 | NaN |

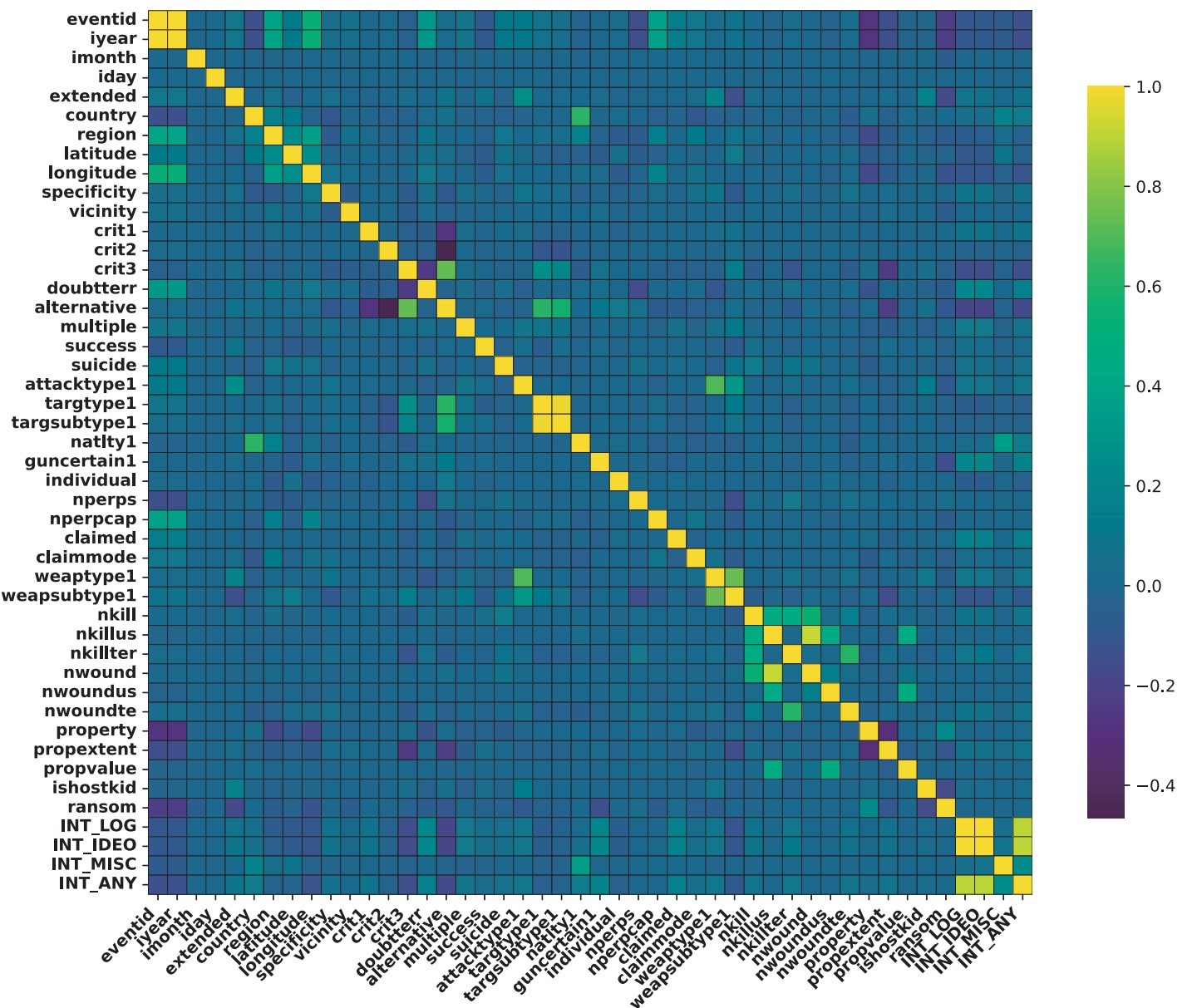

**Figure 8 Correlation between continuous variables.**

These observations underscore intricate relationships within terrorist attack dynamics, crucial for analysis and strategic intervention. NMI visualizes categorical feature correlations in Fig. 9, where darker hues indicate stronger associations. Key patterns include:

- **Associations between Attack Type and Target Type:** Certain attack types target specific entities, reflecting strategic decisions.
- **Associations between Target Nationality and Weapon Type:** Weapon choices correlate with target nationalities, influenced by geopolitical factors.
- **Associations between Perpetrator Nationality and Region:** Perpetrator origins align with attack regions, indicating regional preferences.

## Imputation of missing values based on feature correlations

Upon detailed analysis of the GTD dataset, we identified significant redundancy across various features. This redundancy can be leveraged to effectively address missing values. In the preprocessing stage, we capitalized on this redundancy to impute missing values, details of which are outlined below:

- **Filling in missing values for the exact date of the event:**

In the GTD dataset, when the exact date of an event is unknown, the corresponding variable is recorded as 0. To tackle this issue, we conducted an in-depth analysis and found that approximately 27.76% of the missing values can be filled using two columns: the event number and the approximate date. The event number comprises a 12-digit value, with the first eight digits representing the date in the format "yyyymmdd," and the last four digits indicating the serial number of the day (*e.g.*, 0001, 0002, *etc*.,). By parsing the "dd" field in the event number, we can fill in missing values for some dates. In cases where the "dd" field is zero, we rely on the "Approximate Date" column in the dataset to populate the "Exact Date" column. If a sample does not record an approximate attack date, we fill it with special values that differ from the normal situation. For this study, we assign a value of 0 to these missing dates.

- **Filling in the missing values of weapon subtypes:**

In the GTD dataset, the weapon type and weapon subtype used in each terrorist attack are encoded using specific terms. The missing rates for these two features were calculated individually. The "Weapon Type" column does not have any missing values, while the "Weapon Subtype" column has a missing rate of 11.56%. A detailed analysis was conducted on these two columns, revealing that all missing values for weapon subtypes can be filled using the corresponding weapon type. The weapon types provide a general classification of the weapons used in each event, while the weapon subtypes offer more specific descriptions for most weapon types. Through analysis, it was found that certain weapon types, such as radioactive weapons, do not have corresponding weapon subtypes. For these samples, their weapon type encoding value is added to 50 and filled in the

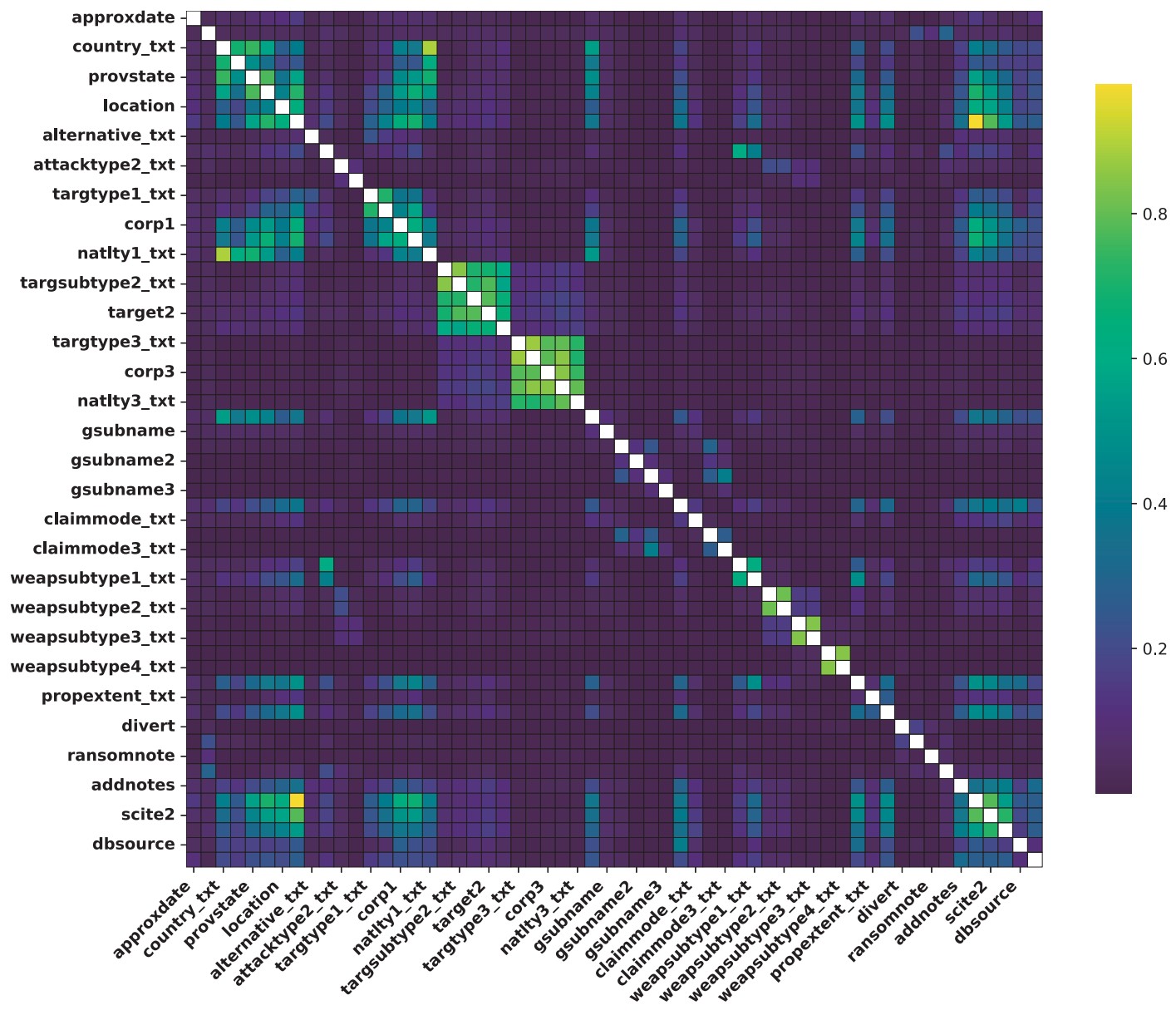

**Figure 9  Correlation between categorical features.**                               

"Weapon Subtype" column. The reason for adding a constant value to the encoding value of the weapon type is to avoid any inconsistencies with the original encoding of the weapon subtype. This addition ensures that there is no conflict with the original encrypted value of the weapon subtype.

- **Filling in the missing values of the number of deaths:**

    The death toll is a crucial measure of the severity of terrorist attacks. In the GTD dataset, the "death toll" column has a missing rate of 4.03%. Through analysis, it was found that

60.05% of the missing data can be addressed by utilizing the effective redundancy of information between the "Number of Deaths in the United States" column, the "Number of Murderer Deaths" column, and the "Number of Deaths" column in the dataset. To fill in the missing values, the data for each sample is examined. If both the "Number of Deaths in the United States" and "Number of Murderer Deaths" columns have values and the "Number of Deaths" column is missing, the available values from the former two columns are combined to assign a value to the number of deaths. If one of the above two columns is missing, the value from the non-missing column is filled into the "Number of Deaths" column. For samples where all three columns are missing, a value of −1 is assigned to the death toll. This facilitates the accurate identification of these samples by the model. To ensure non-negative values in the death count, we incremented the death toll by 1 in all instances. The same approach is applied to filling in missing values in the "Injured" column as in the "Deaths" column.

- Imputation of missing values for the victim's nationality: The nationality of the victim is a crucial aspect recorded and coded in the GTD dataset under the "Nationality of Target/Victim" feature. However, missing values may exist for this feature.

  Several approaches were considered:

- Utilize the location of the attack: If the attack location is known, infer the victim's nationality, assuming they are primarily from the country of the attack.
- Use the nationality of the target: Assume the victims share the nationality of the target.
- Use the perpetrator's nationality: Assume the victims share the nationality of the perpetrator.
- Utilize imputation methods: Use statistical methods to predict missing values by analyzing available data for patterns and relationships.

Through exploratory data analysis, we discovered a strong correlation between the nationality of the victims in certain samples and the country where the attack occurred. To quantify this correlation, we calculate the conditional probability-based correlation, denoted as $\sigma$. The conditional probability-based correlation $\sigma$ is calculated using the formula:

$$\sigma = P(B|A) = \frac{P(AB)}{P(A)} \tag{6}$$

where: $P(AB)$ is the joint probability of events A and B occurring together. $P(A)$ is the probability of event A occurring. This formula helps quantify the correlation between the nationality of the victims and the country where the attack occurred, aiding in the effective imputation of missing values.

## Deletion of redundant features

Deleting redundant features is a critical step in ML model training. Dealing with a large number of features in a dataset can lead to the curse of dimensionality, resulting in

increased model complexity and decreased generalization capability. Therefore, enhancing data processing efficiency and improving model prediction accuracy requires the elimination of invalid redundant features from the GTD dataset. The correlation between features was utilized to fill in missing values. If a feature column was used for filling, and its information was already included in the filled feature columns, then that particular column was eliminated. For instance, once the "Exact Date" column was populated with data from the "Approximate Date" column, the "Approximate Date" column could be safely removed. Similarly, if there were feature columns in the dataset containing identical information, such as the "Country code where the attack occurred" column and the "Country name where the attack occurred" column, one of them, like the "Country name where the attack occurred" column, was removed. This reduction in dimensionality resulted in improved data processing efficiency and enhanced the overall performance of the ML model.

## Generation of derived features

To enhance the distinction between samples and optimize the performance of the ML model, we leverage prior knowledge to generate a range of derived features. These methods can be categorized into three categories: addition of basic features, division of basic features, and addition of binary-encoded basic features.

- Addition of basic features:

   By combining the "number of dead" column and the "number of injured" column, we can generate the total number of casualties caused by terrorist attacks. Similarly, we can calculate various statistics related to the perpetrators of terrorist attacks, such as the number of casualties caused by them.

- Division of basic features:

   To gain further insights, we calculate proportions by dividing specific columns. For instance, dividing the "number of murderer's deaths" column by the "number of deaths" column allows us to determine the proportion of deaths caused by the murderers. It is important to note that we add one to the value in the "Number of Deaths" column to avoid division by zero errors. Similarly, we can calculate proportions related to the number of injuries caused by the perpetrators.

- Addition of binary-encoded basic features:

   In the GTD dataset, information sources for each terrorist attack event are recorded as the first cited source, second cited source, and third cited source. To process this information, we first binary encode each citation source. A value of 1 is assigned if the citation source is present, and 0 is assigned if it is missing. By summing up the binary-encoded features from the three columns, we obtain the count of main information sources used in compiling each terrorist attack event.

Following this selective preprocessing process, the refined dataset now encompasses a total of 180,706 incidents, each characterized by a more streamlined and informative set of 59 features. Data and features determine the upper limit of the ML model. We performed data cleaning, missing value filling, correlation analysis, and other operations on the GTD dataset and finally obtained 59 features. Among them, there are 30 numerical features, 28 categorical features, and only one feature content short text (summary of the attack event). By preprocessing data samples, the noise in the original data can be effectively reduced, the data processing efficiency can be improved, and a solid foundation can be laid for the next step of modelling work.

## Addressing potential biases

The GTD dataset exhibits significant data imbalance, particularly in the distribution of Gname responsible for attacks. This imbalance is evident when examining Table 7, which illustrates the distribution of Gname. Table 7 underscores the varying frequencies of attacks associated with each group. To address potential biases, we implemented two strategies to ensure the data's integrity and representativeness in the GTD. The first strategy was applied to the entire dataset, and the second focused on the Gname attribute.

The first strategy applied during the preprocessing phase includes several steps as follows:

(i) **Balanced imputation approach:**

In dealing with missing values, we employed a balanced imputation approach. For numerical data, we used mean imputation for normally distributed data and median imputation for skewed data to avoid outliers. For categorical data, mode imputation or assigning a distinct 'unknown' category was employed. This approach helps to preserve the original distribution of the data and prevent the introduction of bias that might occur if a disproportionate number of records were filled with a non-representative value.

(ii) **Critical evaluation of filled values:**

We evaluated the impact of filled values by analyzing data distribution before and after imputation to ensure consistent variance and skewness. This step is crucial for maintaining the dataset's statistical properties and ensuring that the models trained on this data do not inherit any systemic bias from the preprocessing phase.

(iii) **Handling redundant features:**

We employed a systematic approach to identify and remove duplicate features. This helped avoid multicollinearity, which can lead to biased or over-fitted models and improved model interpretability and generalizability.

(iv) **Diverse feature selection:**

To further mitigate bias, we ensured a diverse selection of features that comprehensively represented different aspects of the data. This holistic view prevents the model from overly relying on a subset of features that might be biased towards certain patterns or trends.

(v) **Continuous monitoring and validation:**

After preprocessing, we monitored and validated our models for bias by evaluating performance across different data subgroups.

**Table 7 Number of incidents associated with various terrorist organizations.**

| Terrorist organization | Number of incidents |
| --- | --- |
| Unknown | 91,906 |
| Taliban | 11,982 |
| Islamic State of Iraq and the Levant (ISIL) | 7,254 |
| Shining Path (SL) | 4,564 |
| Al-Shabaab | 4,419 |
| New People's Army (NPA) | 3,395 |
| Farabundo Marti National Liberation Front (FMLN) | 3,351 |
| Boko Haram | 3,320 |
| Houthi extremists (Ansar Allah) | 3,196 |
| Irish Republican Army (IRA) | 2,670 |
| Kurdistan Workers' Party (PKK) | 2,582 |
| Revolutionary Armed Forces of Colombia (FARC) | 2,490 |
| Communist Party of India—Maoist (CPI-Maoist) | 2,093 |
| Maoists | 2,091 |
| Basque Fatherland and Freedom (ETA) | 2,024 |
| National Liberation Army of Colombia (ELN) | 1,815 |
| Liberation Tigers of Tamil Eelam (LTTE) | 1,602 |
| Tehrik-i-Taliban Pakistan (TTP) | 1,490 |
| Palestinians | 1,123 |
| Al-Qaida in the Arabian Peninsula (AQAP) | 1,113 |
| Fulani extremists | 1,099 |
| Separatists | 927 |
| Muslim extremists | 924 |
| Nicaraguan Democratic Force (FDN) | 895 |
| Manuel Rodriguez Patriotic Front (FPMR) | 830 |
| Sikh Extremists | 716 |
| Corsican National Liberation Front (FLNC) | 641 |
| Al-Qaida in Iraq | 638 |
| Donetsk People's Republic | 636 |
| Khorasan Chapter of the Islamic State | 614 |
| African National Congress (South Africa) | 607 |
| Abu Sayyaf Group (ASG) | 591 |
| Palestinian Extremists | 583 |
| Sinai Province of the Islamic State | 558 |
| Tupac Amaru Revolutionary Movement (MRTA) | 557 |
| M-19 (Movement of April 19) | 555 |
| Bangsamoro Islamic Freedom Movement (BIFM) | 538 |

## (vi) Comprehensive sampling

The second strategy, implemented specifically on the Gname attribute, aimed to rectify the skewed distribution of incidents among the selected organizations. To achieve this, the

study utilized SMOTE, a sophisticated resampling method tailored to address class imbalances (*Ahmed, Hameed & Bawany, 2022*). Dealing with imbalanced datasets commonly involves oversampling and undersampling methods to alter the data distribution. However, undersampling methods may result in the loss of existing sample information, while oversampling methods can lead to classifier overfitting. SMOTE-T combines oversampling and undersampling with Tomek links to reduce sample overlap. SMOTE generates new minority class samples to balance the dataset, and noisy samples are removed. The SMOTE algorithm enhances the representation of minority classes by generating synthetic samples. It operates by randomly selecting a sample from the minority class and performing linear interpolation between this sample and its $m$ nearest neighbors within the same class. The algorithm's basic steps are as follows:

1) Identify the $m$ nearest minority class neighbors for a given minority class sample $x_i$.
2) Randomly select one of these $m$ neighbors, denoted as $x_j$.
3) Generate a new sample through random linear interpolation between $x_i$ and $x_j$, adding it to the minority class.

For handling samples with noise, particularly when two samples $x_i$ and $x_j$ from different classes are nearest neighbors to each other, they form a Tomek link pair. The presence of Tomek links indicates overlapping class boundaries or noise. Removing such links, specifically the majority class instance in each pair, makes the dataset's class boundaries more distinct, enhancing the classification algorithm's performance (*Sasada et al., 2020*). SMOTE generation of synthetic samples aims to achieve a more balanced class distribution, addressing the issue of under-representation. The addition of Tomek links helps to refine the dataset further by removing overlapping samples between classes, thereby improving the robustness and quality of the data for training models. This dual approach ensures a more balanced representation of classes and enhances the dataset's cleanliness, contributing to more accurate and reliable predictive modeling. Setting a threshold of 500 incidents, we focus on terrorist organizations with significant involvement in the GTD, ensuring a substantial dataset for in-depth analysis and efficient model training. This criterion enhances predictive accuracy by concentrating on groups with enough data to establish reliable patterns and trends, while reducing noise from sporadic activities. Despite this, further refinement is needed to address the remaining imbalance in the dataset.

To address the skewed distribution of incidents, we employ SMOTE-T, a resampling method that generates synthetic samples for underrepresented classes and uses Tomek links to clean overlapping samples. This dual strategy improves the balance and quality of the dataset, enhancing model generalization and robustness. Algorithm 1 outlines the procedure for implementing SMOTE-T, addressing the residual imbalance and improving predictive modeling capabilities across different terrorist organizations.

---

**Algorithm 1   Pseudo-code for data balancing for Gname.**

1: **Input:** Dataset $D$, Threshold $\theta = 500$

2: **Output:** Balanced Dataset $D_{balanced}$

3: **Preprocessing Step:**

4: Exclude groups with 'unknown' names and those with incidents $< \theta$.

5: Retain groups with significant impact, based on the incident count.

6: **Step 1: Initialization**

7: Initialize $D_{balanced} \leftarrow \varnothing$

8: Identify relevant classes in $D$ based on $\theta$

9: **Step 2: Balancing using SMOTE-T**

10: **for** each class $y$ in relevant classes **do**

11:     Extract subset $D_y$ for class $y$

12:     **if** $|D_y| < \theta$ **then**

13:         Apply SMOTE to $D_y$ to generate $D_{y_{synth}}$

14:         $D_{balanced} \leftarrow D_{balanced} \cup D_{y_{synth}}$

15:     **else**

16:         $D_{balanced} \leftarrow D_{balanced} \cup D_y$

17:     **end if**

18: **end for**

19: **Step 3: Refinement using Tomek Links**

20: Apply Tomek Links to $D_{balanced}$ to remove overlaps

21: Update $D_{balanced}$ after refinement

22: **return** $D_{balanced}$ as the final solution

---

## Feature extraction with transformer-based DistilBERT

In this study, we employed the DistilBERT model for text embedding extraction. DistilBERT is a more efficient and deployable version of the BERT architecture, designed for real-world applications (*Ghanadian, Nejadgholi & Al Osman, 2024*; *Karande et al., 2021*). It retains most of BERT's performance while being smaller and faster, making it advantageous for processing extensive textual data, such as that found in terrorism databases (*Shahinmoghadam, Kahou & Motamedi, 2024*). The model's reduced complexity allows for quicker analysis without significant loss of context or accuracy. DistilBERT is particularly effective in real-time threat detection, as it can detect subtle nuances in text, crucial for accurately classifying complex terrorist event data.

This investigation capitalizes on DistilBERT for deriving text embeddings through the fine-tuning of this pre-trained model across various epochs. The model's architecture, which maintains the efficiency and speed of BERT while minimizing its size, proves advantageous for analyzing extensive textual datasets common in terrorism research.

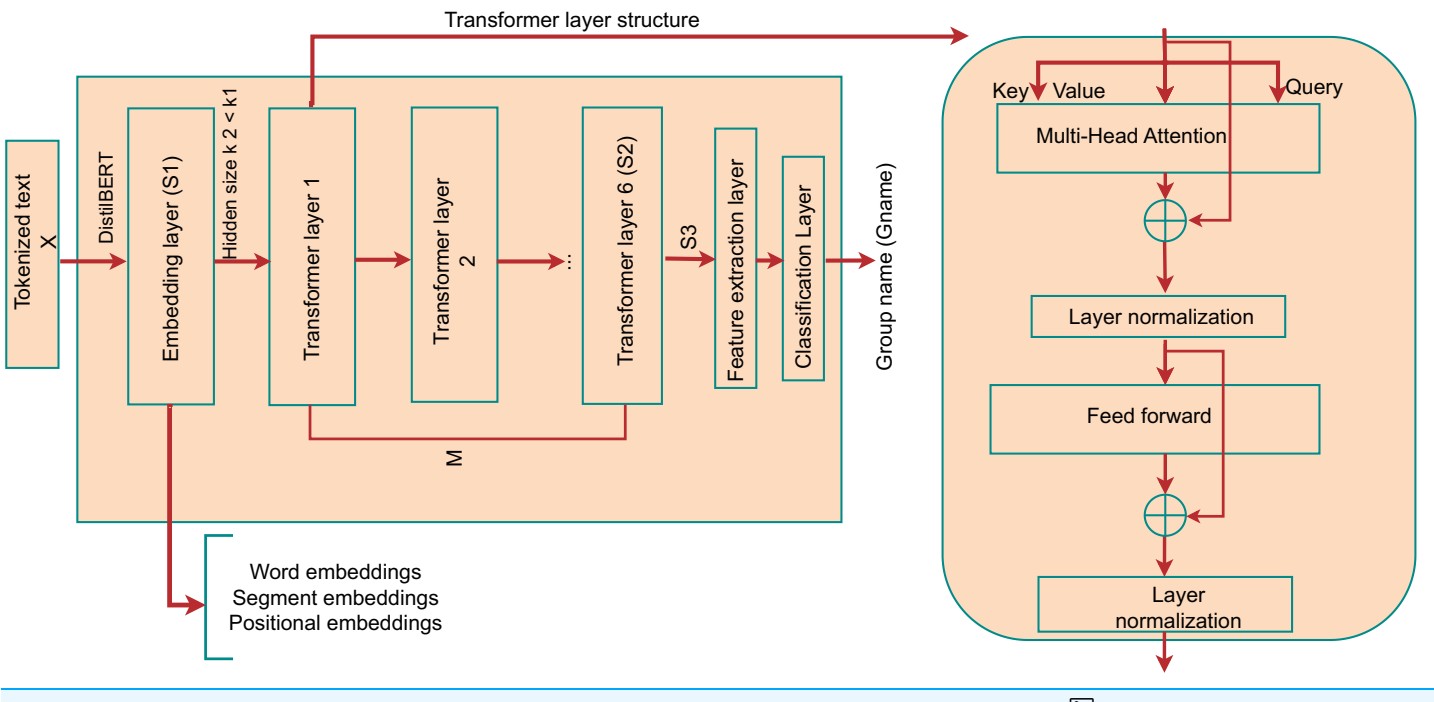

**Figure 10 Architecture of the DistilBERT model.**               

The ability of DistilBERT to expedite analytical processes without compromising on contextual depth or accuracy is depicted in Fig. 10, highlighting its foundation on the BERT base model (*Bert-base-uncased*) and delineating key distinctions from the original BERT framework:

i)   A reduction in parameters by 40%, making DistilBERT leaner than its BERT counterpart.

ii)  Enhancement of inference speed by 60%

iii) Adoption of dynamic rather than static masking for inference.

iv)  Exclusion of next sentence prediction (NSP) and segment embedding in the training stage.

v)   Incorporation of six transformer layers, as opposed to the twelve found in the BERT base.

vi)  Reduction in training duration to 3.5 GPU days from the 12 required by the original model.

DistilBERT undergoes training using the same datasets as the BERT base, encompassing the Toronto Books *Corpus* and English Wikipedia (*Oralbekova et al., 2023*).

As illustrated in Fig. 10, modifications include substituting the original classification layer in DistilBERT with two separate layers designated for feature extraction and Gname classification. The model processes input sentences X, denoted as a series of tokens $X = x_1, \ldots, x_s$, to yield a unified semantic vector [CLS] per sentence. Prior to token sequence processing, DistilBERT employs sub-word and word-piece tokenization

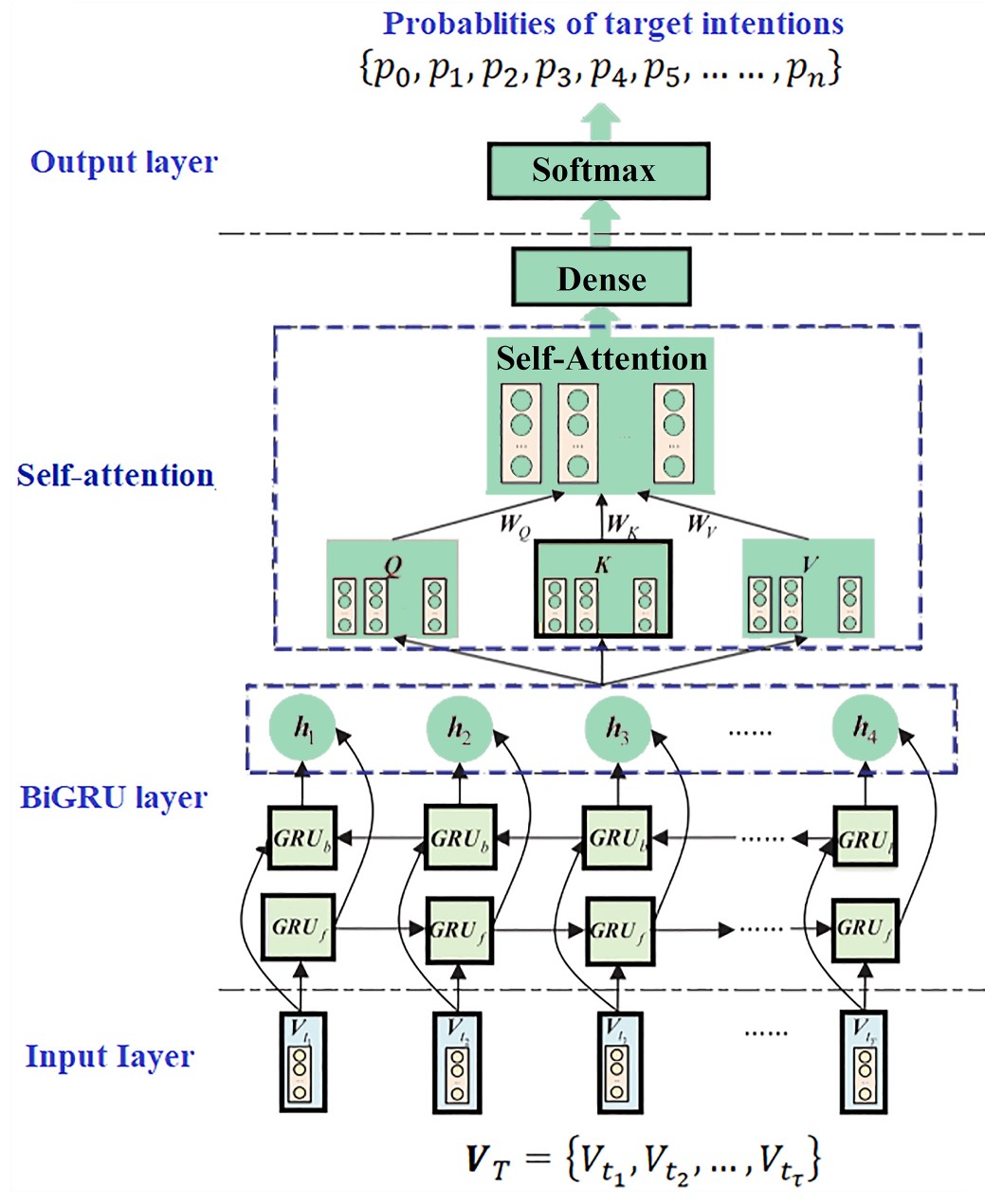

**Figure 11 The BiGRU-SA model for classifying and predicting terrorist organizations.**

techniques (*Kowsher et al., 2022*) to generate input embedding vectors (S1). This tokenization assigns each token a trio of embeddings: word, segment, and positional, with special tokens [SEP] and [CLS] marking the beginning and end of sequences. Subsequently, the multi-layered RNN, leveraging a self-attention mechanism known as the lexicon encoder, aggregates these embeddings to form a cohesive contextual vector S2. The semantic embeddings S3 are derived by amalgamating the contextual vectors S2, orchestrated by the [CLS] token.

Feature extraction is facilitated through a fully connected layer, processing `[CLS]` token embeddings to yield a vector $F$ with dimension $d = 128$, thereby reducing the dimensionality from the original 768 to 128. These condensed features, $d = 128$, are subsequently integrated with pertinent features for classification purposes, employing the GELU activation function (*Xiong et al., 2023*) for enhanced model efficacy, as formulated in Eq. (7).

$$\text{GELU}(m) = m \cdot \Phi(m) \tag{7}$$

where, $m$ symbolizes the fully connected layer's output, with $\Phi(m)$ representing the Gaussian distribution's cumulative distribution function.

The concluding classification phase involves refining the model and adjusting the pre-trained weights $W$ for the Gname identification task, framed as a multi-class classification challenge. This phase features a fully connected layer with $r = 2$ neurons, indicative of class count, where the Softmax function assigns probabilities to class $c$ memberships for input $X$. Training of the classification layer is executed through the cross-entropy loss function.

## BiGRU-SA model

This model is structured into four layers: the input layer, the BiGRU layer, the self-attention layer, and the output layer, as shown in Fig. 11. Each layer plays a critical role in analyzing and inferring the intentions and characteristics of terrorist organizations based on the attack data recorded in the GTD.

The input layer serves as the initial stage, where relevant data from the GTD is fed into the model. The BiGRU layer, utilizing bidirectional gated recurrent units, processes the sequential information in both forward and backward directions, capturing dependencies from past and future states. Following this, the self-attention layer allows the model to weigh the importance of different parts of the input data and capture long-range dependencies within the sequence. Finally, the output layer produces the predicted classifications of terrorist organizations responsible for the recorded attacks in the GTD based on the processed information from the previous layers. These layers integrate to form an advanced architecture designed to effectively capture and interpret the complex temporal dynamics inherent in terrorist activity data. Details concerning each layer, their respective functions, and how they are integrated are presented as follows:

    1. **Input layer**

The input layer employs different strategies to preprocess the collected data and transform it into a feature vector suitable for direct processing by the subsequent BiGRU layer augmented with an attention mechanism.

- DistilBERT features strategy: This strategy processes the textual features extracted by DistilBERT from the summary text. These features are encoded using DistilBERT embeddings to capture nuanced semantic information. Let $T$ be the summary text and $D(T)$ be the DistilBERT embeddings of $T$:

$$\mathbf{E}_T = D(T) \tag{8}$$
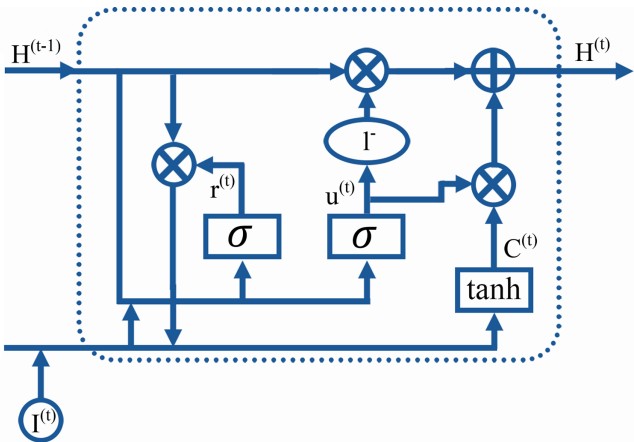

**Figure 12 Gated recurrent unit structure.**

- This strategy integrates both textual and non-textual features. Non-textual features are first converted into textual format and then concatenated with the summary text. This combined text is then fed into DistilBERT. Let $T$ be the summary text, $N$ be the non-textual features converted into textual format, and $D(\cdot)$ represent DistilBERT embeddings:

$$\mathbf{E}_{T+N} = D(T \oplus N) \tag{9}$$

where $\odot$ denotes the concatenation of $T$ and $N$.

- Combined features strategy: In this strategy, individual features are combined after encoding categorical features through label encoding. This converts categorical data into numerical format, which is then merged with the textual features extracted by DistilBERT into a unified vector representation. Let $C$ be the categorical features, $L(C)$ be the label-encoded categorical features, and $X$ be the numerical features. Let $T$ be the summary text and $D(T)$ be the DistilBERT embeddings of $T$:

$$\mathbf{L}_C = L(C) \tag{10}$$

Then, combine the numerical features $X$, the encoded categorical features $\mathbf{L}_C$, and the DistilBERT embeddings $\mathbf{E}_T$:

$$\mathbf{F}_{\text{combined}} = \mathbf{E}_T \oplus \mathbf{L}_C \oplus X \tag{11}$$

where $\mathbf{F}_{\text{combined}}$ is the unified vector representation combining numerical, textual, and encoded categorical data. The input layer orchestrates these diverse feature types into a coherent characteristic vector for effective processing by the subsequent BiGRU layer augmented with an attention mechanism.

   2. **Gated recurrent unit network:**
   The GRU network is an evolution of the long short-term memory (LSTM) model that simplifies the architecture by merging the input and forget gates into a single update gate $u(t)$. This gate determines how much of the past state information should be passed to the

current state. Additionally, a reset gate $r(t)$ decides how much of the past state information should be discarded. The architecture of the unit is depicted in Fig. 12. Where, $I(t)$ denotes the input at time $t$, while $C(t)$ and $H(t)$ are the candidate and actual hidden states at time $t$, respectively. The symbols $\Theta_u, \Theta_r, \Theta_c$ are the weight matrices associated with the update gate, reset gate, and candidate state, and $b_u, b_r, b_c$ are their respective biases. The functions $\sigma$ and tanh are the activation functions used, and the symbol $\odot$ indicates element-wise multiplication.

The sequence of operations for computing the forward pass in a GRU is outlined below:

$$u^{(t)} = \sigma\left(\Theta_u\left[I^{(t)}, H^{(t-1)}\right] + b_u\right)$$
$$r^{(t)} = \sigma\left(\Theta_r\left[I^{(t)}, H^{(t-1)}\right] + b_r\right)$$
$$C^{(t)} = \tanh\left(\Theta_c\left[I^{(t)}, r^{(t)} \odot H^{(t-1)}\right] + b_c\right)$$
$$H^{(t)} = u^{(t)} \odot H^{(t-1)} + \left[1 - u^{(t)}\right] \odot C^{(t)}$$

This formulation captures the essence of GRU's operation, focusing on the update and reset mechanisms that govern the flow and transformation of information through the network over time. However, the GRU has a limitation in that it only considers historical information before the current moment and ignores any influence from future moments. This unidirectional nature hinders the model's ability to capture the complete context of the input sequence.

To address this limitation, we can utilize the bidirectional GRU (BiGRU) structure. BiGRU is composed of two GRU networks: a forward GRU and a backward GRU. The forward GRU processes the input sequence in the normal chronological order, while the backward GRU processes the input sequence in the reverse order. This bidirectional propagation mechanism allows the output node of BiGRU at each time step to contain both historical and future information about the current moment in the input sequence.

Considering both past and future information allows BiGRU to better capture the association between forward and backward characteristics in the time-series data. This capability enables the model to process contextual information more effectively, leading to improved network performance.

In the context of predicting terrorist organizations in the GTD dataset, BiGRU can benefit from its bidirectional nature to capture both past and future events related to terrorist activities. This enhanced understanding of the temporal context can help the model uncover hidden patterns and dependencies, ultimately improving its predictive capabilities for identifying and analyzing terrorist organizations.

As depicted in Fig. 13, the hidden layer state $h_t$ at time $t$ for the BiGRU is a combination of both forward $\vec{h}_t$ and backward $\overleftarrow{h}_t$ states. The forward state $\vec{h}t$ is determined by the current input $x_t$ and the previous state $ht - 1$, whereas the backward state $\overleftarrow{h}_t$ is influenced by $x_t$ and the subsequent state $ht + 1$. This illustrates the bidirectional nature of the GRU, encapsulating past and future information for a complete temporal insight. The BiGRU
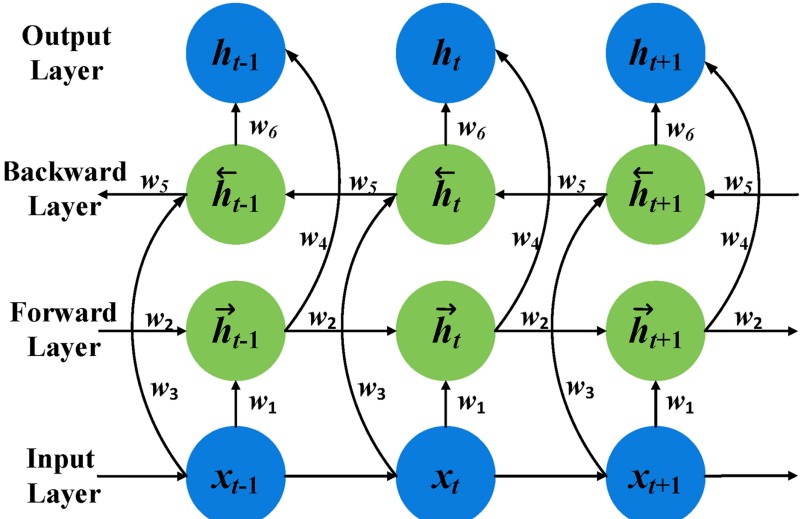

**Figure 13 Bidirectional gated recurrent unit structure.**

computational approach, detailed in Eq. (12), systematically combines inputs from both past and future, constructing hidden layer states at each timestep to enable a detailed analysis of temporal sequences. This method underscores the BiGRU's distinctive ability to gather and integrate temporal data from two directions, thus improving sequence modeling.

For the forward pass:

$$\overrightarrow{h_t} = \sigma(w_1\, X_t + w_2\, \overrightarrow{h}_{t-1})$$
$$\overleftarrow{h_t} = \sigma(w_3\, X_t + w_4\, \overleftarrow{h}_{t+1})$$
$$X'_t = \phi(w_5\, \overrightarrow{h}_t + w_6\, \overleftarrow{h}_t) \tag{12}$$

Here, $W_i$ (for $i = 1$ to 6) represent the weights transitioning between layers, while $\sigma$ and $\phi$ denote distinct activation functions. This formulation ensures a nuanced capture of information from both past and future contexts, pivotal for sequence analysis in BiGRU frameworks.

3. **Self-attention layer**

The self-attention mechanism, also known as the scaled dot-product attention, is a technique that can be integrated with the BiGRU model to enhance its performance in capturing important information from the input sequence (*Zhang et al., 2018*; *Kenarang, Farahani & Manthouri, 2022*). The self-attention mechanism allows the model to focus on different parts of the input sequence while making predictions, enabling it to selectively attend to the most relevant features.

In our case, we implement the self-attention mechanism by introducing an attention layer between the BiGRU layer and the final output layer. This attention layer takes the hidden states from the BiGRU layer as input and computes a set of attention weights for

each hidden state. These attention weights represent the importance or relevance of each hidden state in the context of the entire input sequence.

The attention weights are typically computed using a scoring function. One common approach is the dot-product scoring function, where the similarity between the hidden states and a context vector is measured by taking the dot product between them. The resulting scores are then passed through a softmax function to obtain the attention weights, ensuring that they sum to 1.

Once we have the attention weights, we compute a weighted sum of the hidden states using these weights. This weighted sum represents the attended representation of the input sequence, where the model has focused more on the important parts of the sequence. This attended representation is then passed through the final output layer to make predictions for our task of analyzing terrorist organization features.

Integrating the self-attention mechanism with the BiGRU model allows the model to dynamically and adaptively focus on the most relevant parts of the input sequence. The self-attention mechanism enables the model to capture long-range interdependencies and internal correlations within the time-series data. Incorporating the self-attention layer into the architecture allows the model to assign higher weights to important characteristics output from the BiGRU network layer. This helps the model learn the relevance and importance of each characteristic and automatically assign corresponding weights during the training process.

For instance, in predicting the likelihood of a terrorist organization's success based on various features such as region, country, attack type, and weapon type, the self-attention mechanism can identify the most influential features and assign higher weights to them. This allows the model to focus more on the relevant features while downplaying the impact of less relevant ones. Attending to the key characteristics helps the model better understand the underlying patterns and associations, improving its predictive capabilities.

The integration of the self-attention mechanism with the BiGRU model helps the model effectively filter out unnecessary details and concentrate on crucial information for predicting terrorist organization features. Enhancing the model's memory capacity and capturing critical dependencies, the self-attention mechanism contributes to improved performance and accuracy in predicting and analyzing characteristics of terrorist organizations in the GTD dataset.

4. **Output layer:**

The output layer processes information from previous layers and applies an activation function to generate final predictions or probabilities for the specified task. Typically, this layer includes one or more fully connected (dense) layers, followed by an activation function that formats the output appropriately, such as mapping it to a range or format that represents the probabilities of different classes. When the task involves classifying features of terrorist organizations, the output layer aims to produce outputs suitable for multi-class classification, categorizing organizations into various predefined groups. This functionality is achieved through the use of the Softmax Regression function, a generalization of logistic regression for multi-class classification. According to Fig. 14, the

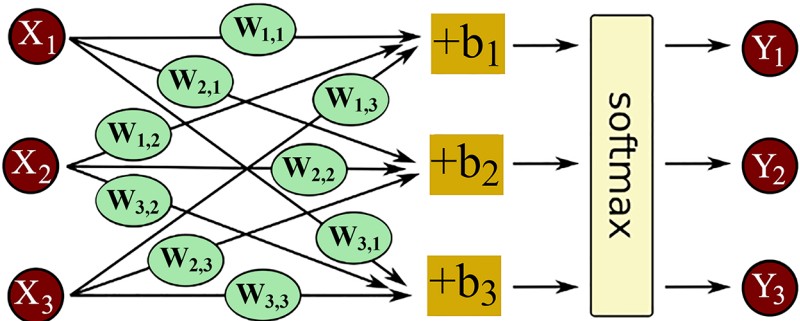

**Figure 14 Softmax regression layer.**

softmax function assigns probabilities to different classes. This makes it ideal for the final layer in neural network architectures designed for classification.

The operation of softmax regression consists of two main steps. First, it computes the likelihood (or evidence) that the input features belong to each class. This is done by calculating a weighted sum of the input features plus a bias term, referred to as the net input $z$, as shown in Fig. 14.

Softmax regression distinguishes itself by using the softmax function $\phi(\cdot)$ instead of the sigmoid function used in logistic regression. The probability that a particular training example $\mathbf{x}^{(i)}$ is assigned to class $j$, given the net input vector $z^{(i)}$ for all classes, is given by:

$$P\left(y = j | z^{(i)}\right) = \phi\left(z^{(i)}\right) = \frac{e^{z^{(i)}}}{\sum\limits_{j=1}^{k} e^{z_j^{(i)}}} \tag{13}$$

where, $z$ is the net input representing the weighted sum of features $x_1, x_2, \ldots, x_m$ plus a bias term $b$, expressed as:

$$z = w_1 x_1 + w_2 x_2 + \ldots + w_m x_m + b = \sum_{l=1}^{m} w_l x_l + b = \mathbf{w}^T \mathbf{x} + b \tag{14}$$

In this context, $\mathbf{w}$ is the weight vector, $\mathbf{x}$ is the feature vector of a single training example, and $b$ is the bias. Each weight $w_l$ corresponds to a feature $x_l$, with $m$ being the total number of features. The softmax function calculates the probability that the training example $\mathbf{x}^{(i)}$ belongs to class $l$, based on the weight vector $\mathbf{w}$ and the net input $z^{(i)}$. For each class label $j = 1, \ldots, K$, it computes the probability $P(y = j | \mathbf{x}^{(i)}; \mathbf{w}_j)$ by normalizing the exponential of the net input for each class against the sum of exponentials for all class net inputs. This ensures that the sum of probabilities for all classes equals one.

For output computation, a linear combination of input variables $x_s$, along with a bias term, is used to determine the inputs to the softmax function, as shown below:

$$\begin{bmatrix} y_1 \\ y_2 \\ y_3 \end{bmatrix} = \text{softmax} \begin{vmatrix} W_{1,1} x_1 + W_{1,2} x_2 + W_{1,3} x_3 + b_1 \\ W_{2,1} x_1 + W_{2,2} x_2 + W_{2,3} x_3 + b_2 \\ W_{3,1} x_1 + W_{3,2} x_2 + W_{3,3} x_3 + b_3 \end{vmatrix} \tag{15}$$

This calculation is further optimized through matrix operations for enhanced computational efficiency:

$$\begin{bmatrix} y_1 \\ y_2 \\ y_3 \end{bmatrix} = \text{softmax}\left( \begin{bmatrix} W_{1,1} & W_{1,2} & W_{1,3} \\ W_{2,1} & W_{2,2} & W_{2,3} \\ W_{3,1} & W_{3,2} & W_{3,3} \end{bmatrix} \cdot \begin{bmatrix} x_1 \\ x_2 \\ x_3 \end{bmatrix} + \begin{bmatrix} b_1 \\ b_2 \\ b_3 \end{bmatrix} \right) \tag{16}$$

Applying the softmax function transforms the computed evidence into class probabilities. Given a net input vector $z$, the softmax function normalizes the exponential values of the net inputs for all classes, ensuring the probabilities sum to one. This normalization allows the model to interpret the outputs as class probabilities and make predictions on the most probable class for each training sample.

# EXPERIMENTS

## Evaluation metrics

To assess the performance of the proposed method for predicting terrorist organizations, several evaluation metrics were employed to provide a comprehensive understanding of the model's generalization ability. The chosen metrics include accuracy (ACC), precision (Precision), sensitivity (Sn), specificity (Sp), and Matthew's correlation coefficient (MCC). These metrics provide a detailed view of the model's predictive capabilities (*Zhuo et al., 2020*).

In the context of classifying and predicting the group responsible for terrorism attacks in the GTD dataset, Sn is crucial as it measures the model's ability to correctly identify instances where the predicted group is indeed responsible for the attack. This is particularly important for effective counterterrorism measures. Sp assesses the model's capacity to avoid falsely attributing attacks to certain groups, which is essential for maintaining the credibility and effectiveness of the classification system (*Ul Qamar et al., 2024*). While ACC is widely used to measure the overall correctness of the model's predictions, it might be misleading in cases of class imbalance, common in terrorism datasets. In such scenarios, MCC provides a balanced assessment by considering both sensitivity and specificity, offering a nuanced evaluation of the model's performance (*Wang et al., 2020*).

$$ACC = \frac{TP + TN}{TP + TN + FP + FN} \tag{17}$$

$$Precision = \frac{TP}{TP + FP} \tag{18}$$

$$Sn = \frac{TP}{TP + FN} \tag{19}$$

$$Sp = \frac{TN}{TN + FP} \tag{20}$$

$$MCC = \frac{TP \times TN - FP \times FN}{\sqrt{(TP + FP)(TP + FN)(TN + FP)(TN + FN)}} \tag{21}$$

where TP represents true positives, TN represents true negatives, FP represents false positives, and FN represents false negatives. By considering these diverse metrics, we gain a

comprehensive understanding of how well our model performs across various aspects of prediction, providing valuable insights into its overall effectiveness.

### Experimental parameter settings

Hyperparameters significantly affect the predictive performance of neural network models. Three-fold cross-validation with random search was used to determine the optimal hyperparameters (*Cheng, Wang & He, 2021*; *Aliper et al., 2016*; *Rimal & Sharma, 2023*). During training, the batch size was selected from [16, 32, 64, 128, 256], the number of iterations from [10, 20, 30, 50, 100], the dropout probability from [0.1, 0.3, 0.5, 0.7], and the learning rate from [0.0001, 0.001, 0.01]. The optimized parameters for the BiGRU-SA model include a batch size of 64, 100 iterations, a dropout rate of 0.5, and a learning rate of 0.01. We conducted a comprehensive comparative analysis to assess the effectiveness of the proposed BiGRU-SA model relative to a range of both traditional and advanced machine learning classifiers. The machine learning models selected for this study included bagging, extreme gradient boosting (XGBoost), random forest (RandomForest), extra trees (ExtraTrees), decision tree (DecisionTree), and light gradient boosting machine (LightGBM) (*González et al., 2020*; *Ashraf et al., 2022*). Default parameters recommended by scikit-learn (sklearn) for multi-classification problems were applied (*Terol et al., 2020*; *Komer, Bergstra & Eliasmith, 2019*). Using default parameters ensures robust initial performance and stability across various datasets, serving as a reliable benchmark for comparative analysis (*Wu et al., 2023*). The deep learning models included in this study were deep neural networks (DNN), GRU, BiGRU, BiGRU-SA. The DNN architecture comprises four fully connected layers with neuron counts decreasing from 256 to 32, employing the Rectified Linear Unit (ReLU) activation function in each hidden layer (*Singh & Sabrol, 2021*). This comprehensive selection of classifiers, spanning both traditional ML techniques and advanced DL architectures, provides a solid benchmark for evaluating the capabilities of the BiGRU-SA model in classifying and predicting terrorist organization activities.

## RESULTS, ANALYSIS, AND DISCUSSION

### Experimental series 1: textual features extracted by DistilBERT

The primary objective of Experimental Series 1, as shown in Table 8, is to assess the performance of various classifiers in identifying types of Gname, focusing on features extracted through DistilBERT. Classifiers were trained and tested using textual features obtained through DistilBERT embeddings. The evaluation includes computing performance metrics such as accuracy, precision, recall, specificity, and MCC to gauge the models' classification accuracy and predictive capability.

### Experimental series 2: all features to text

Experimental Series 2, as shown in Table 9, investigates the impact of concatenating categorical and numerical features with textual features extracted from the GTD dataset. The main goal is to explore how combining different types of features influences the classification and prediction of Gname. Classifiers were trained and tested using the

**Table 8 Textual features only extracted by DistilBERT.**

| Classifier | ACC (%) | Precision (%) | Sn (%) | Sp (%) | MCC |
|---|---|---|---|---|---|
| Bagging | 85.60 | 85.99 | 84.26 | 88.93 | 0.7830 |
| XGBoost | 87.42 | 86.03 | 82.33 | 91.52 | 0.8231 |
| RandomForest | 75.15 | 73.47 | 78.37 | 71.93 | 0.5243 |
| ExtraTrees | 88.59 | 88.67 | 82.98 | 93.20 | 0.8375 |
| DecisionTree | 89.47 | 88.89 | 83.99 | 96.95 | 0.8548 |
| LightGBM | 89.75 | 89.87 | 84.44 | 95.05 | 0.8399 |
| DNN | 90.56 | 88.48 | 86.49 | 90.62 | 0.8643 |
| GRU | 91.87 | 89.41 | 87.62 | 96.13 | 0.8708 |
| BiGRU | 92.71 | 89.81 | 89.00 | 97.42 | 0.8968 |
| BiGRU-SA | **93.68** | **90.06** | **90.83** | **98.68** | **0.9150** |

**Table 9 All features to text.**

| Classifier | ACC (%) | Precision (%) | Sn (%) | Sp (%) | MCC |
|---|---|---|---|---|---|
| Bagging | 88.60 | 88.99 | 86.26 | 90.93 | 0.803 |
| XGBoost | 90.42 | 89.03 | 85.33 | 93.52 | 0.853 |
| RandomForest | 77.15 | 75.47 | 80.37 | 73.93 | 0.564 |
| ExtraTrees | 91.59 | 91.67 | 85.98 | 95.20 | 0.857 |
| DecisionTree | 92.47 | 91.89 | 86.99 | 97.95 | 0.874 |
| LightGBM | 92.75 | 92.87 | 87.44 | 96.05 | 0.859 |
| DNN | 93.56 | 91.48 | 89.49 | 93.62 | 0.884 |
| GRU | 94.87 | 92.41 | 90.62 | 98.13 | 0.900 |
| BiGRU | 95.71 | 92.81 | 92.00 | 98.42 | 0.916 |
| BiGRU-SA | **95.80** | **93.06** | **93.83** | **99.68** | **0.935** |

combined features obtained through concatenation. Similar to Series 1, the evaluation process involves computing performance metrics to assess the effectiveness of the concatenated features in improving the classification and prediction of Gname.

## Experimental series 3: individual features combination

The third experimental series, as shown in Table 10, explores the combination of features extracted from each individual feature set after conversion. In this series, each feature set is converted individually into a suitable format for processing. Once converted, the features are combined to form a comprehensive feature set for analysis and classification. This series aims to assess the effectiveness of combining features derived from different aspects of the data. By converting and combining each feature set separately, we aim to capture the unique information each feature type provides and evaluate how their combination influences the classification performance.

| Table 10 Combination features based on individual. | | | | | |
|---|---|---|---|---|---|
| **Classifier** | **ACC** (%) | **Precision** (%) | **Sn** (%) | **Sp** (%) | **MCC** |
| Decision trees | 95.71 | 92.82 | 92.50 | 98.00 | 0.917 |
| Bagging | 93.20 | 93.22 | 90.50 | 97.50 | 0.895 |
| RandomForest | 95.86 | 94.22 | 92.96 | 98.20 | 0.930 |
| ExtraTrees | 95.95 | 94.22 | 92.62 | 98.50 | 0.927 |
| XGBoost | 95.16 | 93.72 | 93.49 | 96.70 | 0.920 |
| LightGBM | 93.75 | 93.87 | 93.90 | 97.00 | 0.905 |
| DNN | 94.56 | 92.48 | 94.00 | 95.00 | 0.900 |
| GRU | 94.87 | 94.91 | 95.20 | 97.80 | 0.920 |
| BiGRU | 96.82 | 95.77 | 93.41 | 99.00 | 0.947 |
| BiGRU-SA | **98.68** | **96.06** | **96.83** | **99.50** | **0.975** |

## RESULT ANALYSIS

The analysis and discussion of the experimental series shed light on the performance of different models and feature combinations for identifying Gname. Through the exploration of three distinct series, each employing varied feature sets and classification methodologies, we glean insights into the effectiveness of different approaches in handling the complex task of Gname classification. According to the figures, connected graphs (curves) depict the performance metrics of each algorithm across the three experimental series. Each curve within a figure illustrates the performance trajectory of a specific algorithm using five metrics: ACC, precision, recall (sensitivity or Sn), Sp, and MCC for all series. In-depth analysis, supported by observations from the figures and tables, highlights the robust capabilities of our algorithms in classifying Gname across various experimental series. The graphical representation of these performances offers a detailed narrative on how each model capitalizes on the features provided. Looking at Fig. 15A, the bagging algorithm shows consistent improvement when transitioning from text-only features in Table 8 to a more diverse set in Table 9. The upward trajectory of its performance curves clearly indicates an enhanced capability to assimilate additional information for more accurate predictions. In Fig. 15B, XGBoost demonstrates a commendable increment in performance metrics as we move through the series. It is particularly responsive to the inclusion of categorical and numerical features, as its precision curve sharply rises, a fact that is well-supported by the higher scores in Table 9. Moving to the RandomForest model, as depicted in Fig. 15C, while showing a moderate climb in performance, hints at limitations within the model's framework to fully exploit the DistilBERT-extracted features. However, the curve does progress positively with integrating additional features, as seen in the corresponding tables. ExtraTrees, illustrated in Fig. 15D, paints an impressive picture of robustness. Its curves, particularly for MCC, show an upward trend, reflecting its strength in balancing true positives and negatives, which is corroborated by the high scores in Table 10.

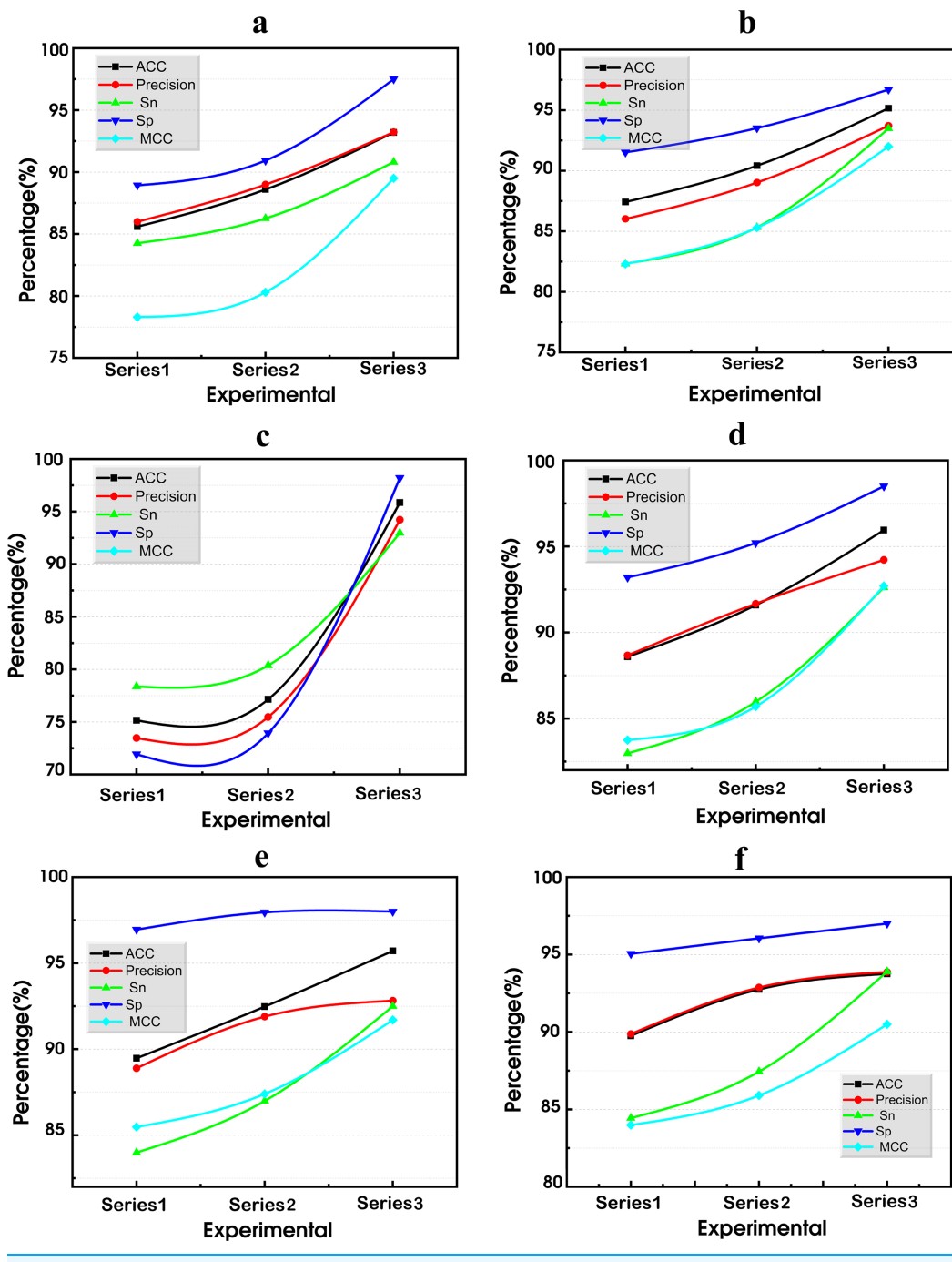

**Figure 15 Comparative analysis of algorithm performance across all series.**

Figure 15E shows the DecisionTree algorithm with a notable performance leap, especially in specificity and MCC, when enriched features are introduced. This visual climb aligns with the jump in scores from Tables 8 to 9. LightGBM, shown in Fig. 15F, exhibits a remarkable ascendancy in curves across all metrics. Its ability to exploit the combined feature set is evident from the consistent elevation across the series, revealing the model's

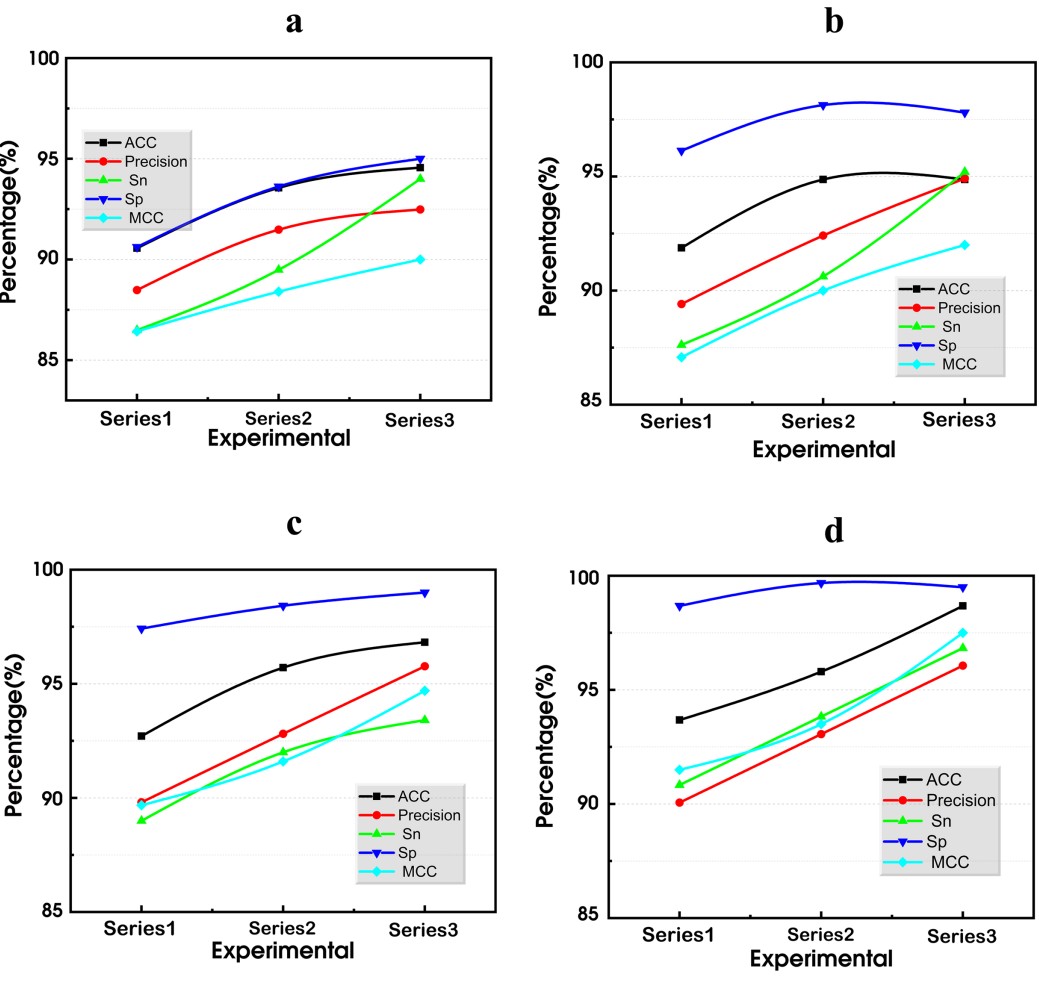

**Figure 16 Comparative performance analysis of DL algorithms across all series.**

adaptability and precision. The curves for DL models in Fig. 16 the DNN, in Fig. 16A, demonstrate a steady and substantial rise in performance, particularly when it comes to capturing the nuanced interplay between textual and non-textual features, as evidenced by its impressive scores across the tables. GRU's performance curve in Fig. 16B underscores its capability to model sequential data effectively. The upward trends across the series mirror the substantial gains reflected in the corresponding tables, with significant jumps in recall and specificity. BiGRU, as shown in Fig. 16C, is harnessed from a bidirectional understanding of the context. This is especially pronounced in Fig. 16C, where individual feature sets are combined, highlighting the algorithm's strength in temporal dependency capture. Lastly, BiGRU-SA's curves in Fig. 16D stand out with the highest elevation across all metrics, affirming the model's exceptional performance noted in the tables. The self-attention mechanism allows for an enhanced focus on pertinent features, contributing to significant strides in accuracy and MCC. In synthesizing these observations, the connected curves reveal a narrative of progressive model improvement and underscore the imperative of feature selection.

As we blend textual with categorical and numerical data, we unlock new levels of classifier proficiency. The self-attention models, such as BiGRU-SA, set a benchmark in effectively navigating the complexities of the GTD dataset, as they dynamically focus on the most salient features to predict Gname with high accuracy and reliability.

A comprehensive evaluation against diverse classifiers, including traditional ML and DL models, illuminates the BiGRU-SA model's relative strengths. Its exceptional performance across various metrics highlights its potential for accurately predicting and classifying terrorist organisations. This emphasises the critical role of feature engineering and model selection in the classification of terrorism-related Gname. Leveraging a combination of textual, categorical, and numerical features and suitable classifiers facilitates achieving high accuracy and precision in identifying Gname from complex datasets like the GTD. Additionally, incorporating advanced architectures such as BiGRU-SA further boosts performance by effectively capturing temporal dependencies and intricate data patterns.

## Generalization ability of the method

This section evaluates how effectively the BiGRU-SA method generalises across various terrorist activity environments, which vary in attack frequencies and levels of organisational involvement. The analysis is grounded in the performance metrics outlined in "Data Preprocessing", with the addition of the F1-score as a crucial metric. The F1-score, being the harmonic mean of precision and recall, adeptly balances the accuracy and completeness of the model's predictions, making it especially valuable in contexts where the implications of false positives and negatives are significant for security measures.

The BiGRU-SA framework's adaptability and effectiveness shine through, consistently outperforming other ML algorithms across all evaluated metrics and datasets in Table 11. The study evaluates the algorithm's performance depicted in the curves from Figs. 17A to 17E.

In Fig. 17A, where the attack range is greater than or equal to 1,000 with 19 terrorist organizations responsible for 50,200 attacks, we would expect to see high performance across algorithms due to the large number of attacks, which likely results in a rich dataset. BiGRU-SA, in particular, would display a near-perfect curve, illustrating its proficiency in handling dense data with complex patterns, supported by the highest accuracy and precision metrics. Moving to Fig. 17B with greater than or equal to 500 attacks by 36 organizations, the performance curves might show a slight dip compared to Case 1, as the data becomes less concentrated.

Algorithms like BiGRU-SA and random forests would still maintain a high level of accuracy, demonstrating their robustness to variations in data volume and distribution. Moving to Fig. 17C, with greater than or equal to 100 attacks from 122 organizations, the curves would start to show more variation. The increased number of organizations might introduce more noise and variability into the data, explaining why algorithms like decision trees and bagging might display more significant dips in their performance curves. Despite this, BiGRU-SA would continue to maintain a relatively high performance due to its ability to capture and prioritize relevant features. Moreover, as shown from Fig. 17D which represents an even broader range of greater than or equal to 50 attacks across 210

**Table 11 Performance of the proposed framework with various algorithms across attack ranges.**

| | Terrorist attacks (range) | ≥1,000 | ≥500 | ≥100 | ≥50 | ≥5 |
|---|---|---|---|---|---|---|
| | Terrorist organisations | 19 | 36 | 122 | 210 | 936 |
| | Total number of terrorist attacks | 50,200 | 58,520 | 78,107 | 84,339 | 94,871 |
| Accuracy (%) | Decision trees | 0.9600 | 0.9571 | 0.9025 | 0.8902 | 0.8608 |
| | Bagging | 0.9608 | 0.9320 | 0.8333 | 0.7999 | 0.7406 |
| | Random forests | 0.9831 | 0.9682 | 0.9045 | 0.8812 | 0.8357 |
| | ExtraTrees | 0.9793 | 0.9595 | 0.8867 | 0.8603 | 0.8032 |
| | XGBoost | 0.9835 | 0.9516 | 0.8539 | 0.7914 | 0.7986 |
| | LightGBM | 0.9425 | 0.9375 | 0.9250 | 0.9125 | 0.8300 |
| | DNN | 0.9500 | 0.9456 | 0.9310 | 0.9180 | 0.8450 |
| | GRU | 0.9550 | 0.9487 | 0.9360 | 0.9230 | 0.8500 |
| | BiGRU | 0.9827 | 0.9586 | 0.8784 | 0.8546 | 0.7962 |
| | BiGRU-SA | 0.9910 | 0.9868 | 0.9440 | 0.9310 | 0.8880 |
| Precision (%) | Decision trees | 0.9770 | 0.9282 | 0.8875 | 0.8456 | 0.6788 |
| | Bagging | 0.9460 | 0.9322 | 0.8713 | 0.7856 | 0.6471 |
| | Random forests | 0.9792 | 0.9577 | 0.9276 | 0.9174 | 0.8207 |
| | ExtraTrees | 0.9733 | 0.9422 | 0.9112 | 0.8613 | 0.7768 |
| | XGBoost | 0.9784 | 0.9372 | 0.8522 | 0.8235 | 0.8269 |
| | LightGBM | 0.9435 | 0.9387 | 0.9260 | 0.9130 | 0.8905 |
| | DNN | 0.9510 | 0.9248 | 0.9320 | 0.9190 | 0.8955 |
| | GRU | 0.9560 | 0.9491 | 0.9370 | 0.9240 | 0.9005 |
| | BiGRU | 0.9733 | 0.9422 | 0.8812 | 0.8613 | 0.7768 |
| | BiGRU-SA | 0.9800 | 0.9606 | 0.9402 | 0.9240 | 0.9005 |
| Recall (%) | Decision trees | 0.9595 | 0.9250 | 0.8905 | 0.8904 | 0.8900 |
| | Bagging | 0.9401 | 0.9050 | 0.8051 | 0.7233 | 0.6048 |
| | Random forests | 0.9747 | 0.9341 | 0.8853 | 0.8396 | 0.8120 |
| | ExtraTrees | 0.9700 | 0.9262 | 0.8615 | 0.8086 | 0.7694 |
| | XGBoost | 0.9761 | 0.9349 | 0.8465 | 0.8379 | 0.8387 |
| | LightGBM | 0.9420 | 0.9390 | 0.9255 | 0.9120 | 0.8895 |
| | DNN | 0.9495 | 0.9400 | 0.9315 | 0.9175 | 0.8945 |
| | GRU | 0.9545 | 0.9520 | 0.8950 | 0.9225 | 0.8995 |
| | BiGRU | 0.9761 | 0.9296 | 0.8860 | 0.8379 | 0.8122 |
| | BiGRU-SA | 0.9750 | 0.9683 | 0.9365 | 0.9295 | 0.9075 |
| F1-score (%) | Decision trees | 0.9682 | 0.9265 | 0.8865 | 0.8867 | 0.7809 |
| | Bagging | 0.9430 | 0.8942 | 0.8361 | 0.7522 | 0.6247 |
| | Random forests | 0.9769 | 0.9459 | 0.9145 | 0.8764 | 0.8163 |
| | ExtraTrees | 0.9715 | 0.9340 | 0.8851 | 0.8341 | 0.7731 |

(Continued)

| | | | | | |
|---|---|---|---|---|---|
| XGBoost | 0.9772 | 0.9361 | 0.8489 | 0.8302 | 0.8328 |
| LightGBM | 0.9427 | 0.9388 | 0.9257 | 0.9125 | 0.8900 |
| DNN | 0.9503 | 0.9324 | 0.9318 | 0.9183 | 0.8950 |
| GRU | 0.9551 | 0.9505 | 0.8865 | 0.8740 | 0.9000 |
| BiGRU | 0.9747 | 0.9359 | 0.8976 | 0.8496 | 0.7935 |
| BiGRU-SA | 0.9775 | 0.9644 | 0.9367 | 0.9233 | 0.9142 |

organizations. The performance curves would further spread, with algorithms like XGBoost and ExtraTrees showing steeper declines, as indicated by their lower accuracy and precision. This case illustrates the challenges posed by more distributed datasets with less frequent attacks per organization, making patterns harder to discern.

According to in Fig. 17E, depicting greater than or equal to five attacks by a vast number of 936 organizations, we observed the most significant dips in performance curves, especially for models like bagging and decision trees. However, BiGRU-SA's curve would remain the highest, although with a noticeable decline compared to previous cases. The increased complexity and noise in the data with many low-frequency attack instances make it challenging for algorithms to maintain high performance. In scenarios with fewer terrorist organizations, as indicated by the performance curves in Fig. 17A through Fig. 17E, the data distribution is less sparse, which considerably benefits models like BiGRU-SA. This allows the model to identify representative patterns with greater accuracy. As such, the accuracy, precision, sensitivity, and F1-scores are notably higher. The improved signal-to-noise ratio in these cases aids the model in distinguishing relevant patterns from irrelevant data effectively, leading to enhanced predictive performance. Moreover, the simpler structure of these datasets enables the model to generalize better across various situations, capturing the underlying trends and dynamics with efficiency. The strength of the BiGRU-SA model is particularly evident in its generalization capabilities, ensuring that it maintains robust performance in diverse settings of terrorist activities. As datasets expand to include a larger number of terrorist organizations, and as the frequency of attacks per organization diminishes, a decline in the performance of all algorithms is observed. Sophisticated models, such as BiGRU-SA, which incorporate self-attention mechanisms, exhibit considerable resilience in the face of these challenges. Moreover, their ability to focus on the most salient features amid data noise and to manage the complexities and imbalances of the datasets becomes apparent. This capability is especially valuable as datasets grow more complex, underscoring the critical role of algorithm selection and its contribution to handling intricate classification tasks.

In the second scenario, the study evaluates the proposed framework's performance using a 10-fold cross-validation approach alongside the hold-out method. Using the hold-out method, the model's performance is assessed on data it has not encountered during the training phase. This approach offers a more accurate estimate of the model's behaviour on

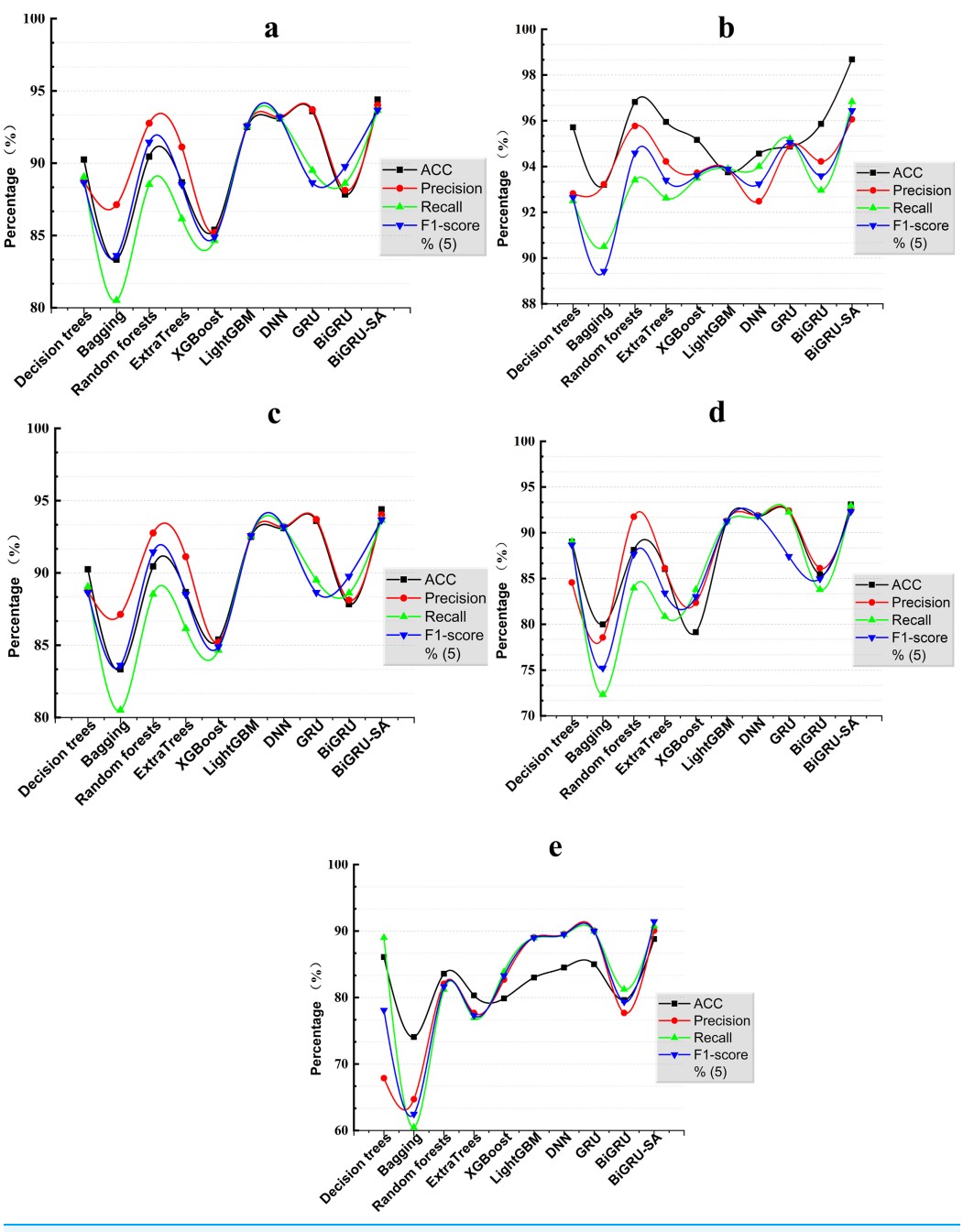

**Figure 17 Comparative analysis of the generalization ability.**

new, unseen data. The hold-out method is widely utilised for model selection, hyperparameter tuning, and comparing the performance of various models. Additionally, this method gauges the model's resilience and efficiency in addressing data imbalances, a common challenge in this field.

Table 12 further explores the performance of various models under two data split verification methods in scenarios with attack frequencies over 500, offering a detailed

**Table 12 Framework performance with ten fold, hold-out methods and (terror attack frequency ≥ 500).**

| | | Data split verification method | | | |
|---|---|---|---|---|---|
| | | Hold-out method | 10-fold cross-validation method | | |
| Metrics | Models | | Mean | Max | Min |
| Accuracy (%) | Decision trees | 0.9586 | 0.9560 | 0.9644 | 0.9499 |
| | Bagging | 0.9320 | 0.9335 | 0.9368 | 0.9276 |
| | Random forests | 0.9682 | 0.9660 | 0.9687 | 0.9626 |
| | ExtraTrees | 0.9595 | 0.9594 | 0.9624 | 0.9560 |
| | XGBoost | 0.9716 | 0.9678 | 0.9708 | 0.9632 |
| | LightGBM | 0.9375 | 0.9375 | 0.9563 | 0.9188 |
| | DNN | 0.9456 | 0.9456 | 0.9645 | 0.9267 |
| | GRU | 0.9487 | 0.9487 | 0.9677 | 0.9297 |
| | BiGRU | 0.9571 | 0.9571 | 0.9763 | 0.9380 |
| | BiGRU-SA | 0.9868 | 0.9743 | 0.9860 | 0.9766 |
| Precision (%) | Decision trees | 0.9281 | 0.9281 | 0.9467 | 0.9095 |
| | Bagging | 0.9322 | 0.9272 | 0.9381 | 0.9116 |
| | Random forests | 0.9577 | 0.9556 | 0.9638 | 0.9478 |
| | ExtraTrees | 0.9422 | 0.9415 | 0.9457 | 0.9352 |
| | XGBoost | 0.9572 | 0.9528 | 0.9568 | 0.9440 |
| | LightGBM | 0.9387 | 0.9387 | 0.9575 | 0.9200 |
| | DNN | 0.9248 | 0.9248 | 0.9433 | 0.9063 |
| | GRU | 0.9241 | 0.9241 | 0.9426 | 0.9056 |
| | BiGRU | 0.9282 | 0.9291 | 0.9436 | 0.9181 |
| | BiGRU-SA | 0.9606 | 0.9556 | 0.9554 | 0.9446 |
| Recall (%) | Decision trees | 0.9250 | 0.9250 | 0.9435 | 0.9065 |
| | Bagging | 0.8581 | 0.8625 | 0.8712 | 0.8550 |
| | Random forests | 0.9341 | 0.9350 | 0.9414 | 0.9290 |
| | ExtraTrees | 0.9262 | 0.9289 | 0.9348 | 0.9235 |
| | XGBoost | 0.9449 | 0.9423 | 0.9505 | 0.9337 |
| | LightGBM | 0.8900 | 0.8900 | 0.9078 | 0.8722 |
| | DNN | 0.9100 | 0.9100 | 0.9282 | 0.8918 |
| | GRU | 0.9120 | 0.9120 | 0.9302 | 0.8938 |
| | BiGRU | 0.9296 | 0.9318 | 0.9441 | 0.9231 |
| | BiGRU-SA | 0.9550 | 0.9575 | 0.9620 | 0.9456 |
| F1-score (%) | Decision trees | 0.9240 | 0.9240 | 0.9425 | 0.9055 |
| | Bagging | 0.8752 | 0.8759 | 0.8862 | 0.8668 |
| | Random forests | 0.9429 | 0.9427 | 0.9494 | 0.9357 |
| | ExtraTrees | 0.9328 | 0.9336 | 0.9374 | 0.9271 |
| | XGBoost | 0.9500 | 0.9465 | 0.9531 | 0.9375 |
| | LightGBM | 0.9108 | 0.9108 | 0.9290 | 0.89.26 |
| | DNN | 0.9173 | 0.9173 | 0.9356 | 0.8990 |
| | GRU | 0.9180 | 0.9180 | 0.9364 | 0.8996 |
| | BiGRU | 0.9286 | 0.9301 | 0.9437 | 0.9206 |
| | BiGRU-SA | 0.9627 | 0.9502 | 0.9664 | 0.9483 |

comparison to understand the framework's adaptability and reliability in handling complex datasets. The 10-fold cross-validation approach is a rigorous evaluation method that simulates real-world data conditions by repeatedly splitting the available data into training and testing sets.

This process ensures the model is not simply memorising specific patterns in the training data but can genuinely learn and generalise to unseen instances. When applied to the BiGRU-SA method, the 10-fold cross-validation results solidify its potential as a valuable tool for counter-terrorism efforts.

Firstly, the BiGRU-SA method consistently outperforms other models across various evaluation metrics in both the hold-out and 10-fold cross-validation approaches (as shown in Table 12). This indicates its ability to learn robust patterns even with data fragmentation introduced by cross-validation. This is particularly impressive considering the inherent challenges of data imbalance in this domain, where the number of attacks from specific organisations might be significantly lower than others.

Secondly, focusing specifically on the 10-fold cross-validation scenario, we observe that the BiGRU-SA method maintains high accuracy, precision, recall, and F1-scores across all folds. This consistency further emphasises its ability to handle data imbalance and effectively generalise to unseen data points. The 10-fold process presents the model with various challenges through cross-validation. It throws a series of data subsets at the model, testing its ability to adapt and learn from each unique combination. The BiGRU-SA method consistently overcomes these challenges, showcasing its remarkable resilience and generalizability. These findings suggest that the proposed framework with the BiGRU-SA method is a high-performing model in controlled settings and a practical and reliable tool for real-world applications. Its ability to handle data imbalance and generalise across diverse scenarios, as demonstrated by the 10-fold cross-validation results, makes it a valuable asset in the fight against terrorism.

## Comparison with existing approaches

In this section, the study comprehensively compares our framework with the BiGRU-SA model against existing methodologies as highlighted in studies (*ALfatih, Li & Saadalla, 2019*; *Bangerter et al., 2022*; *Talreja et al., 2017*). This comparison showcases our framework's advancements over traditional ML techniques, referenced in Table 13. Through performance metrics analysis for varying terrorist attack scenarios and terrorism organisation numbers, the proposed framework demonstrates superior accuracy and adaptability, especially in addressing data imbalances. This comparative discussion reveals our model's enhanced capability to predict and analyse terrorist activities more precisely, contributing significantly to counter-terrorism research. The comparison primarily focuses on accuracy and F1-score. The accuracy pinpoints the overall correctness of predictions, gauging the model's general classification ability.

In counter-terrorism scenarios, both accuracy and identifying true positives hold immense significance. F1-score balances precision (identifying true positives) and recall (not missing true positives), offering a more nuanced performance view, especially critical for imbalanced datasets like GTD. False positives can lead to severe repercussions while

**Table 13 Comparison of prediction with state-of-arts.**

| Study | Acc | F1-Score | Number of terrorist organizations |
|---|---|---|---|
| *ALfatih, Li & Saadalla (2019)* | 89.00 | 82.00 | 20 groups |
| *Bangerter et al. (2022)* | 87.81 | 84.41 | 20 groups |
| *Talreja et al. (2017)* | 87.00 | 85.50 | (Not mentioned) |
| Proposed model | 98.60 | 96.27 | 36 groups |

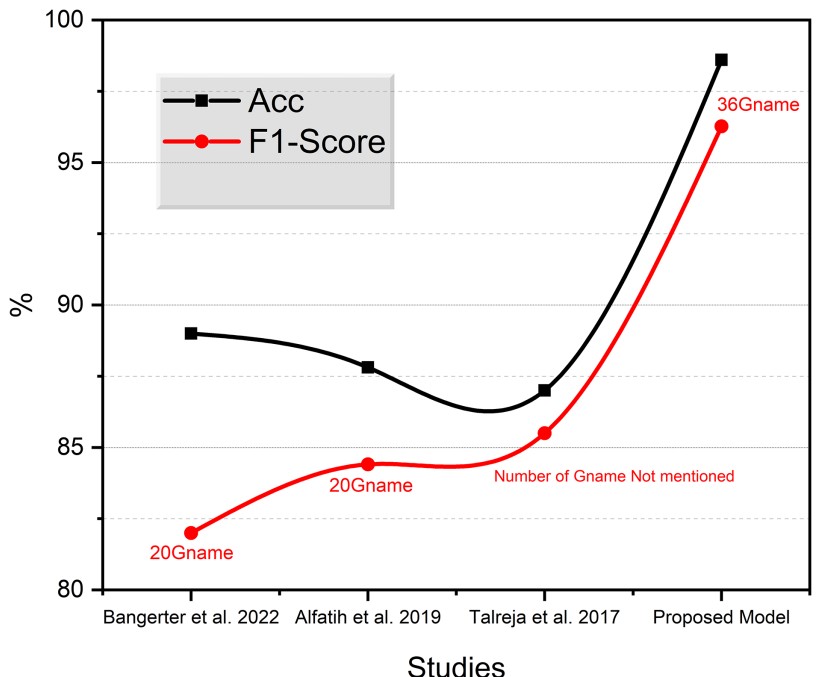

**Figure 18 Comparison of study performances with the state-of-the-art.**

missing true positives hampers prevention efforts. As shown in Table 13 the proposed model exhibits a marked improvement in accuracy and F1-score, achieving 98.6% and 96.27% respectively, surpassing other cited studies. Furthermore, the proposed model considers more terrorist organizations (36 groups) than the 20 groups analyzed in other studies, suggesting a broader application range. The higher accuracy and F1-score, combined with a broader scope of terrorist organization classification, highlight the model's superior performance. As presented in Fig. 18 this significant improvement implies that our model generalises terrorist organisations better and correctly classifies them while minimising false positives. Furthermore, including 36 organisations in our assessment adds another validation layer to our model's robustness. By successfully handling a more extensive and potentially more intricate dataset with potential data sparsity issues, our model demonstrates its broader applicability and adaptability to complex scenarios.

## CONCLUSION

Terrorist organizations pose a significant threat to global security, making the prediction of their activities based on historical data a complex task. However, recent advancements in DL models, particularly in NLP and time series analysis, offer promising avenues for improvement. In this study, we proposed a novel framework centred around a bi-directional gated recurrent unit (BiGRU) and a self-attention mechanism for classifying and predicting terrorist organizations. By leveraging textual features extracted from DistilBERT and addressing data imbalance with the SMOTE-T method. The proposed framework achieved remarkable performance metrics, with the BiGRU-SA model attaining an accuracy of 98.68%, precision of 96.06%, sensitivity of 96.83%, specificity of 99.50%, and Matthews correlation coefficient of 97.50, respectively. These results underscore the efficacy of our model in classifying and predicting terrorist organizations with high reliability and precision. Furthermore, we assessed the effectiveness of several classifiers and observed that both BiGRU-SA and BiGRU consistently delivered superior prediction accuracies. Additionally, the random forest algorithm demonstrated robust classification performance across various metrics. This methodology holds potential for extending its applicability to a wider array of terrorist organizations. The proposed model contributes to the prediction of global terrorist attacks, identifies relevant factors, and provides decision support for anti-terrorism organizations and related countries. With joint efforts and advanced models based on AI, we anticipate that these technologies will enhance the accuracy of predictions related to terrorist groups, thereby bolstering efforts and proactive measures to combat terrorism. A significant limitation of our study is the reliance of the model's effectiveness on the continual updating and refinement of the dataset. The dynamic nature of global terrorism, characterized by the emergence of new groups, the evolution of existing ones, and the changing methods of attack, necessitates a dataset that is regularly updated to accurately reflect these changes. Without timely updates, the model may lose its predictive accuracy over time as it becomes outdated relative to the current state of global terrorism. This limitation opens several avenues for future work, including developing automated or semi-automated processes for data collection and updates, which could ensure that the dataset remains current thereby preserving the model's relevance and accuracy. Such processes could leverage advancements in web scraping, NLP, and ML to identify and integrate new data from a variety of sources, including news reports, government releases, and specialized counter-terrorism databases.

## NOMENCLATURE

The most commonly used abbreviations in this article are summarized in below:

**GTD**     Global Terrorism Database
**ML**      Machine Learning
**DL**      Deep Learning
**BNN**     Bayesian Neural Network
**RELU**    Rectified Linear Unit

| MLP | Multilayer Perceptron |
| BAAD | Broad Agency Announcement Database |
| EDTG | Extended Data on Terrorist Groups |
| RNN | Recurrent Neural Network |
| DGAN | Deep Generative Adversarial Network |
| T2A | Tweet-to-Act |
| Gname | Terrorism Group name responsible for attacks in GTD |
| AI | Artificial Intelligence |
| NLP | Natural Language Processing |
| GRU | Gated Recurrent Unit |
| SA | Sentiment Analysis |
| BiGRU | Bidirectional Gated Recurrent Unit |
| SOMTE | Synthetic Minority Over-sampling |
| EDA | Exploratory Data Analysis |
| PCC | Pearson Correlation Coefficient |
| NMI | Normalized Mutual Information |
| BLSTM | Bidirectional Long Short-Term Memory |

### Funding

The authors received no funding for this work.

### Competing Interests

Natalia Kryvinska is an Academic Editor for PeerJ.

### Author Contributions

- Mohammed Abdalsalam conceived and designed the experiments, performed the experiments, analyzed the data, performed the computation work, prepared figures and/or tables, authored or reviewed drafts of the article, and approved the final draft.
- Chunlin Li performed the experiments, authored or reviewed drafts of the article, and approved the final draft.
- Abdelghani Dahou conceived and designed the experiments, performed the experiments, authored or reviewed drafts of the article, and approved the final draft.
- Natalia Kryvinska performed the experiments, analyzed the data, authored or reviewed drafts of the article, and approved the final draft.

### Data Availability

The train and test datasets are available at Figshare:

ABDALSALAM, MOHAMMED (2024). Test Data. figshare. Dataset. https://doi.org/10.6084/m9.figshare.26129887.v1

ABDALSALAM, MOHAMMED (2024). Training dataset. figshare. Dataset. https://doi.org/10.6084/m9.figshare.26129863.v1

## Supplemental Information

Supplemental information for this article can be found online at http://dx.doi.org/10.7717/peerj-cs.2252#supplemental-information.

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
