# Peer review of "Terrorism group prediction using feature combination and BiGRU with self-attention mechanism"

_PeerJ Computer Science, doi:10.7717/peerj-cs.2252_

## Round 0.1 · original submission · Major Revisions

The manuscript needs major revisions in its presentation, analysis and even in figures and citations. Authors should address these concerns and submit the revised version.

**Language Note:** PeerJ staff have identified that the English language needs to be improved. When you prepare your next revision, please either (i) have a colleague who is proficient in English and familiar with the subject matter review your manuscript, or (ii) contact a professional editing service to review your manuscript. PeerJ can provide language editing services - you can contact us at [email protected] for pricing (be sure to provide your manuscript number and title). – PeerJ Staff

Reviewer 1 ·

Basic reporting

1) The manuscript lacks the inclusion of keywords. Please integrate relevant keywords to enhance the discoverability and indexing of the research paper.
2) The authors must incorporate an abbreviation table into the manuscript that would greatly enhance readability and comprehension for readers.
3) Section 1.2, which discusses research motivation and contribution, appears convoluted and overly lengthy, resulting in potential confusion for readers. The author should consider revising it for clarity and conciseness.
4) Some citation mistakes should be overcome, like “Jiang et al. Jiang et al. (2023) introduced an integrated DL framework designed to incorporate the” in line 187, “a recent study by Ogundunmade et al. Ogundunmade and Adepoju (2024) employed a Bayesian Neural” in line 196, “Another study by Buffa et al. Buffa 200 et al. (2022) investigated the spatial prediction of terrorism across Europe: “ in line 200
5) The Related Work section requires significant improvement. The author should include the contribution, challenges, method, and pros and cons of each research article in the given section in tabular format to enhance the efficiency of the manuscript.
6) Table 1 appears inappropriate; please consider redefining it.
7) The sharpness of Figure. 2, 5, 6, 7, 8, 11, 12, 14 and 18 are not good enough. It is necessary to improve them to make them easy to read.
8) The paper has many grammatical and typographical errors and inconsistent mathematical expressions. They can be easily seen from the text. Read carefully.
9) The author is advised to carefully review the manuscript to identify and address large paragraphs. Dividing lengthy paragraphs into smaller, more digestible ones can improve readability and comprehension. Additionally, removing unnecessary lengthy paragraphs will streamline the content and enhance the overall quality of the manuscript.

Experimental design

The rigorous investigation has been performed for result calculation.

Validity of the findings

The data on which the conclusions are based is robust, statistically sound, and controlled.

Reviewer 2 ·

Basic reporting

I appreciate the efforts of the authors for choosing and analysing this domain.

Presentation & organization of article is good

Experimental design

on what basis incident threshold is choosen in Gname?

The data set allows you to classify the terrorist attack activities. How this analysis helps in predicting the attck activities by terrorist organizations?

can you explain the attack prediction model?

Validity of the findings

while comparing the performance analysis, whether authors considered the optimized models for the other ML & DL models?

Additional comments

on what basis incident threshold is choosen in Gname?

The data set allows you to classify the terrorist attack activities. How this analysis helps in predicting the attck activities by terrorist organizations?

can you explain the attack prediction model?

while comparing the performance analysis, whether authors considered the optimized models for the other ML & DL models?

Reviewer 3 ·

Basic reporting

The paper entitled “Terrorism group prediction using combined features and BiGRU with self-attention mechanism” proposed a framework for classifying and predicting terrorist groups using a Bi-directional Gated Recurrent Unit (BiGRU) enhanced with a self-attention mechanism. It leverages combined features from the Global Terrorism Database, employing advanced Machine Learning and Deep Learning techniques for improved prediction accuracy. The framework, termed BiGRU-SA, demonstrates exceptional efficacy in classifying 36 terrorist organizations and achieved promising results; however, my comments are below:

Introduction:
1. Peerj computer science do not support section numbering, please address the sections numbering.
2. Please check the appropriate citation style for this journal. As far as I know, it is author-year. The manuscript uses only “author (year)” citations throughout, and no parenthetical citations “(author, year).” This format should only be used when the cited work forms part of the text, e.g., “author (year) proposes method X” and the latter otherwise, e.g., “we method differs from method X (author, year).”
3. Figure 1 has no scientific value; I would suggest removing it.
4. I would suggest summarizing the introduction in 3 pages maximum and avoiding the use of subsections.
5. Is better to introduce the “Problem formulation” in a separate section.
6. All the techniques and methods that are not proposed in this work such as GRU, EDA, PCC, etc. it should be cited.
7. The list of contribution should be rewritten, to the best of your understanding, “Pre-processing Policies for Data Integrity, Combination of Relevant Features, and Focused Analysis on High-Impact Groups” are not contributions.
Background and related work:

8. There are citation issues in lines 195, 200, 209, 228, 230, etc.
9. The authors repeatedly introduce the abbreviation, for example, GRU in lines 272 and 1095.

Experimental design

10. The authors introduced the proposed framework in 22 pages. I suggest being more specific about their proposed work and avoiding details of every technique they use (4 – 5 lines with citations of the original paper are enough).
11. Figure 2 is confusing, the data flow should be reviewed.
12. Section 3.1 Dataset: I would suggest moving this subsection to the experiments unless the authors generated the data themselves.
13. Figure 4 has no scientific value; I would suggest removing it.
14. The authors have introduced their work utilizing GRU. Given that Transformer-based models and their subsequent generation currently lead in many NLP tasks as the state-of-the-art, it would be beneficial if the authors elucidated the advantages of GRU over multi-head attention mechanisms.

Validity of the findings

15. I would suggest merging the EXPERIMENTAL SERIES and RESULT ANALYSIS section.

---

## Round 0.2 · accepted · Accept

Revisions carried out by the authors are agreed by the reviewers.

Reviewer 1 ·

Basic reporting

Good

Experimental design

Rigorous results

Validity of the findings

meanigful

Reviewer 3 ·

Basic reporting

I appreciate the authors' efforts in choosing and analyzing this domain. The article's presentation and organization are good.

Experimental design

The investigation was performed well for result calculation.

Validity of the findings

The data on which the conclusions are based are good and statistically sound.